# Spontaneous Transformation of a p53 and Rb-Defective Human Fallopian Tube Epithelial Cell Line after Long Passage with Features of High-Grade Serous Carcinoma

**DOI:** 10.3390/ijms232213843

**Published:** 2022-11-10

**Authors:** Yu-Hsun Chang, Tang-Yuan Chu, Dah-Ching Ding

**Affiliations:** 1Department of Pediatrics, Hualien Tzu Chi Hospital, Buddhist Tzu Chi Medical Foundation, Tzu Chi University, Hualien 97005, Taiwan; 2Department of Obstetrics and Gynecology, Hualien Tzu Chi Hospital, Buddhist Tzu Chi Medical Foundation, Tzu Chi University, Hualien 97005, Taiwan; 3Institute of Medical Sciences, Collagen of Medicine, Tzu Chi University, Hualien 97005, Taiwan

**Keywords:** high-grade serous ovarian cancer, tumorigenicity, transformation, long-term culture, copy number variant

## Abstract

Ovarian cancer is one of the most lethal gynecological cancers, and 80% are high-grade serous carcinomas (HGSOC). Despite advances in chemotherapy and the development of targeted therapies, the survival rate of HGSOC has only moderately improved. Therefore, a cell model that reflects the pathogenesis and clinical characteristics of this disease is urgently needed. We previously developed a human fallopian tube epithelial cell line (FE25) with p53 and Rb deficiencies. After long-term culture in vitro, cells at high-passage numbers showed spontaneous transformation (FE25L). This study aimed to compare FE25 cells cultured in vitro for low (passage 16–31) and high passages (passage 116–139) to determine whether these cells can serve as an ideal cell model of HGSOC. Compared to the cells at low passage, FE25L cells showed increased cell proliferation, clonogenicity, polyploidy, aneuploidy, cell migration, and invasion. They also showed more resistance to chemotherapy and the ability to grow tumors in xenografts. RNA-seq data also showed upregulation of hypoxia, epithelial-mesenchymal transition (EMT), and the NF-κB pathway in FE25L compared to FE25 cells. qRT-PCR confirmed the upregulation of EMT, cytokines, NF-κB, *c-Myc,* and the Wnt/β-catenin pathway. Cross-platform comparability found that FE25L cells could be grouped with the other most likely HGSOC lines, such as TYKNU and COV362. In conclusion, FE25L cells showed more aggressive malignant behavior than FE25 cells and hence might serve as a more suitable model for HGSOC research.

## 1. Introduction

Epithelial ovarian cancer (EOC) ranks fifth among cancer-related deaths in women. Among EOC, high-grade serous ovarian carcinoma (HGSOC) is the leading cause of death [1,2]. More than 220,000 new ovarian cancer cases happened yearly and most patients with ovarian cancer are diagnosed at a late stage of the disease with a 5-year survival rate of 30–40% only [1].

Current treatment options include debulking surgery and adjuvant chemotherapy. Recently, targeted therapy with poly ADP ribose polymerase inhibitors (PARPi) and anti-vascular endothelial growth factor (VEGF) has been used for the treatment of ovarian cancer [3]. Despite the introduction of these novel therapies, the survival rate of ovarian cancer has not improved [1] primarily because ovarian cancer is detected at an advanced stage.

EOC is divided into two types, based on complex pathogenic and molecular pathways [4]. Type 1 EOC includes clear cell, endometrioid, low-grade serous, and mucinous carcinomas. Type 2 ovarian cancers include HGSOC, carcinosarcoma, and undifferentiated carcinoma [4]. A previous study showed that HGSOC originates from the fallopian tube epithelial cells (FTEC) [5]. Genetic changes in HGSOC include extensive structural genomic variations (copy number variation), *p53* pathway inactivation, homologous recombination-mediated DNA damage repair (HR DDR) deficiency, *BRCA1* mutation, cyclin E1 (*CCNE1*), notch receptor 3 (*NOTCH3*) activation, *Rb*, and *NF1* inactivation [4]. These molecular pathways may provide therapeutic opportunities for HGSOC treatment.

Somatic *TP53* mutations are found in more than 96% of HGSOC cases reported in The Cancer Genome Atlas (TCGA) [6]. The *TP53* gene encodes the p53 protein, which has a tumor suppressor function by arresting the cell cycle and activating DNA repair proteins to repair damaged DNA or initiate apoptosis if the damage is too large to be repaired [6]. Approximately half of HGSOCs also harbor somatic or germline mutations or methylation silencing of homologous recombination repair genes, such as BReast Cancer gene 1 (*BRCA1*) or *BRCA2* [6,7]. The other half of HGSOC, called non-BRCAness, typically has *Rb/CCNE1* pathway disruption and *TP53* mutations [8]. When the *p53* and *Rb* pathways are disrupted, the cell cycle typically fails during mitosis, resulting in severe DNA polyploidy and aneuploidy [9].

As FTEC are the cells of origin for HGSOC [10], previous studies proposed that FTEC could be a new model for ovarian serous carcinogenesis [11]. We immortalized FTEC by transfection of human papillomavirus 16 *E6/E7* genes, resulting in the downregulation of *P53* and *Rb* genes [12]. Other transformed FTECs with transfection *RAS*, *c-myc*, and *CCNE1* were also published [13,14]. We hypothesized that after long-term in vitro culturing (passage more than 100 times), the immortalized FTEC (FE25) may undergo spontaneous transformation and show malignant characteristics more like HGSOC.

This study aimed to compare the low passage (passage 16–31) and high passage (passage 116–139, FE25L) versions of FE25 regarding malignant behaviors, which may help reveal the carcinogenesis mechanisms leading to HGSOC.

## 2. Results

### 2.1. FE25L Cells Exhibited Faster Proliferation and More Cells in S and G2/M Phases

To determine the characteristics of FE25L cells, we compared their morphology, proliferation rate, and cell cycle distribution with FE25 cells. FE25L cells exhibited similar morphology to that of FE25 cells, showing a cobblestone-like appearance (Figure 1A). The proliferation rate of FE25L cells was significantly faster than that of FE25 cells (*p* < 0.05, on day 3 and day 5, Figure 1B). Quantification of the proportion of cells in different phases of the cell cycle showed that significantly more FE25L cells were in the S and G2/M phases than that in FE25 cells (30% vs. 21%, *p* < 0.01) (Figure 1C). Moreover, FE25L showed a higher DNA content or ploidy than FE25 cells.

### 2.2. FE25L Had Higher Migration and Invasion Activity Than FE25 Cells

To determine the migration and invasion capabilities of FE25 and FE25L cells, we performed trans-well migration and invasion assays. Migration was significantly higher in the FE25L group than in the FE25 (nearly 9-fold, *p* < 0.01, Figure 2A). In the invasion assay, FE25L cells were found to be more invasive than FE25 cells (nearly 6.5-fold, *p* < 0.01; Figure 2B). These results indicate that FE25L had significantly increased migration and invasion capabilities compared to FE25 cells.

### 2.3. FE25L Formed More Colonies Than FE25 Cells

Clonogenicity activity of the two cells were characterized by colony forming assay on FE25 and FE25L cells and found that there were nearly 7 fold more colonies formed by FE25L than FE25 cells (*p* < 0.001, Figure 3).

### 2.4. FE25L Exhibited Greater Tumorigenesis Potential

Next, we examined the tumorigenic capability of the two cell lines in NSG mice. After subcutaneous injection of 1 × 10^6^ FE25 or FE25L cells into NSG mice, FE25L generated tumors in 3/3 mice on days 165 (1 mouse) and 183 (2 mice). FE25 generated tumors in only one of the three mice on day 173 (Figure 4A). The gross appearance of the xenografts in wax blocks are shown in Figure 4B. Hematoxylin and eosin staining revealed more pleomorphic cells in the FE25 tumor and more epithelioid cells in the FE25L tumor (Figure 4C). Nuclear atypia with significant nuclear pleomorphism with large, bizarre, and multinucleated form was noted in the xenografts.

Immunohistochemistry revealed that both tumors were positive for PAX8, WT1, and CK7 (Figure 4D). Moreover, FE25L tumors showed higher WT1 expression than FE25 (70% vs. 30%, *p* < 0.001, Figure 4E).

### 2.5. FE25L Acquired More Severe Aneuploidy

Chromosomal instability with copy number variation (CNV) is a phenotype characteristic of HGSOC [15]. We analyzed the karyotypes of the FE25 and FE25L cells. We found marked polyploidy and aneuploidy in FE25L cells compared with FE25 cells (Figure 5). Specifically, FE25 cells were sub-diploid with 42 chromosomes at passage 31 (Figure 5A), and FE25L cells were sub-triploid with 59–74 chromosomes at passage 139 (Figure 5B).

### 2.6. Copy Number Variation in FE25L vs. FE25

DNA copy number variations were evaluated by whole-exome sequencing of FE25 (passage 20) and FE25L (passage 124) cells (Figure 6A,B). Compared to normal cells, both FE25 and FE25L showed amplification of driver genes such as *CCNE1* and *NOTCH1*, which were also found in the TCGA-HGSOC study [16]. Other amplified oncogenes frequently found in epithelial ovarian cancer were *ERBB2*, *STAT3*, *PIK3C2B*, *CDK2*, and *CDK4* [16], as well as *SWI/SNF* chromatin modification complex genes, including *ARID1A* and *SMARCC1* [16]. TCGA-confirmed tumor suppressor genes with a lower copy number in the two cells included *RB1* and *CDKN2A/B* [16]. Other copy number reduced genes included *SMAD4*, *HNF1B*, and some gene loci that are not known to be tumor suppressor genes, such as K*RAS*, *MYC*, *PIK3CA*, *and IL6*.

Compared to FE25, FE25L had gain sites at chr2p25, 7q21,22, 8p11, and 20q11, which are related to breast cancer and chronic myelogenous leukemia (Table 1) [17,18]. FE25L loss sites were noted on chromosome 11, related to complement, coagulation, oxidative phosphorylation, estrogen early response, and *mTORC1* signaling (Table 2).

Comparing somatic copy number variation (SNV) to RNA-seq data, three genes showed compatible increased expression and four genes showed decreased expression in FE25L (Figure 6C) and were confirmed by qRT-PCR (Figure 6D). *CASP1* (caspase 1) was found to have decreased expression in FE25L, which correlates with tumorigenesis characteristics.

### 2.7. Differentially Regulated Gene Sets in FE25L Compared to FE25 Cells

The relative increase or decrease in the gene copy number in FE25L does not necessarily result in changes in expression. We performed an RNA-seq array on these two cells to reveal the gene expression changes between FE25L and FE25. Differentially regulated genes in FE25L compared to FE25 were identified and subjected to gene ontology analysis using GSEA [19]. We selected 100 upregulated and 100 downregulated genes (highest fold-change) for the analysis (these genes may be prominent genes in this phenotype). The three most significantly enriched gene sets in upregulated genes included hypoxia, epithelial-mesenchymal transition (EMT) (*VEGFA*, *COL6A2*, *COL6A3*, *ADAM12*, *TFPI2*, *MXRA5*, *LAMA1*, *SERPINE2*, and *SFRP1*), and NF-κB pathways (Table 3). The three most significant enrichment of gene sets in downregulated genes included the *interferon-γ* signaling pathway, interferon-α signaling pathway, and EMT (*TIMP3*, *THBS1*, *COL3A1*, *VCAN*, *COL4A2*, *IGFBP4*, *TAGLN*, *CDH6*, *COL1A2*, *APLP1*, *and NTM*) (Table 4). The above results suggest that FE25L contains more cancer transformation-related genes.

### 2.8. FE25L Exhibited Higher Expression of EMT, Cytokine, NF-κB, and c-myc Signaling Pathways

Next, we explored the differences in the expression of related signaling pathway genes between FE25 and FE25L. TCGA ovarian cancer array analyzed 570 human HGSOC tumors using a whole-genome array. The prognosis of ovarian cancer patients with a high EMT index is worse than that of patients with a low EMT index [20]. We found lower expression of E-cadherin and increased expression of fibronectin, vimentin, and Twist in FE25L cells, which may reflect more EMT in FE25L cells than in FE25 cells (Figure 7A). Cytokines and growth factors secreted by tumor tissues in the ovarian cancer microenvironment play a role in the immune escape, tumor progression, and cancer dissemination [21]. We found higher expression of *interleukin (IL)-6*, *IL-8*, bone morphogenetic protein 2 *(BMP2)*, insulin-like growth factor 2 *(IGF2)*, and *VEGFA* in the FE25L group than in the FE25 group (Figure 7B). *Ccnd1* overexpression is strongly associated with shortened progression-free survival in human ovarian carcinomas [22]. NF-κ*B* is also associated with decreased survival rate in ovarian cancer [23]. Genes involved in the NF-κB pathway include *Ccnd1*, *S100a4*, *Spp1*, and *IkBkε* [24]. We found lower *S100A4* and *SPP1* expression in FE25L than in FE25 (Figure 7C). HGSOC is characterized by numerous copy number alterations, among which overexpression of the myc oncogene occurs in half of the tumors [25]. We found that *c-Myc* expression was higher in FE25L cells than in FE25 cells (Figure 7D). Taken together, FE25L expressed more TCGA HGSOC-related genes than FE25 did.

### 2.9. FE25L Cells Exhibited Increased Wnt/β-Catenin Expression

The Wnt/β-catenin signaling pathway was examined by Western blotting and was used to check protein expression in the cytoplasm and nucleus. The Wnt/β-catenin pathway is associated with cancer stem cell self-renewal, chemoresistance, and metastasis in all subtypes of epithelial ovarian cancer [26]. In our study, β-catenin protein expression was increased in both the cytoplasm and nucleus of FE25L cells (Figure 8A–D). Taken together, FE25L expressed more β-catenin than FE25 did.

### 2.10. FE25L Cells Are More Sensitive to Chemotherapeutic Drugs

We performed an XTT assay to evaluate the proliferation and half-maximal inhibitory concentration (IC_50_) of chemodrugs in FE25 (p20) and FE25L (p125) cells (*n* = 3). The IC50 of taxol for FE25L was significantly lower than FE25 (0.396 ± 0.1 nM vs. 41.6 ± 0.1 nM, *p* < 0.001, Figure 9A,B). The IC_50_ of doxorubicin for FE25L was significantly lower than FE25 (1.4 ± 0.1 μM vs. 3.1 ± 0.1 μM, *p* < 0.001) (Figure 9C,D). The IC_50_ of carboplatin for FE25L was significantly lower than FE25 (2.0 ± 0.1 μM vs. 26.4 ± 0.1 μM, *p* < 0.001) (Figure 9C,D). FE25L cells were found to be more sensitive to chemotherapeutic drugs because more mutations were noted.

### 2.11. Correlation of Gene Expression between FE25 and FE25L Cells and Current HGSOC Lines

A previous study used numerical scores to group current ovarian cancer cell lines into three groups: likely HGSOC (Group 1), probably HGSOC (Group 2), and unlikely HGSOC (Group 3) models of HGSOC [27]. KURAMOCHI or COV362 cell lines belong to Group 1, and SKOV3 or A2780 cells belong to Group 3 [27]. We used genetic features identified in the likely HGSOC cancer cell lines (Group 1) to compare with FE25 and FE25L [27]. Using the top 500 diverse genes of Group 1 HGSOC, FE25 and FE25L cells were grouped with Group 1 cells, including JHOS2, CAOV3, COV362, and TYKNU (Figure 10).

## 3. Discussion

FE25 cells derived from FTEC were immortalized with human papillomavirus with knockdown of *TP53* and *Rb* genes [28]. Several studies using FE25 cells have investigated their roles in ovarian carcinogenesis [29,30]. In previous studies, treatment with ovulatory follicular fluid (FF) or growth factors within FF resulted in more malignant behavior in FE25 cells similar to HGSOC [12,28,31,32]. Given the chromosomal instability due to defective p53 and Rb, we hypothesized that FE25 cells might develop HGSOC phenotype after long-term culture. In this study, we demonstrated that higher passage FE25 cells, FE25L had an increased proliferation, clonogenicity, DNA aneuploidy, migration and invasion capabilities, and higher drug resistance. In terms of gene expression, FE25L showed upregulated EMT, cytokines, NF-κB, c-Myc, and Wnt/β-catenin pathway-associated genes compared to those in FE25 cells. FE25L cells also showed higher tumorigenicity with WT1 expression.

Distinct from human cells needing transgenes (e.g., *hTERT*) to conquer senescence, murine cells are easily immortalized in culture and may transform spontaneously after long-term culture [33]. McCloskey et al. observed spontaneous transformed mouse ovarian surface epithelial cells after a long passage and determined the aberrant Wnt/β-catenin and NF-κB signaling and expression of WT1, PAX8, and cytokeratin mimicking human HGSOC [24]. Another murine background cell line, ID8, is derived from the ovarian surface epithelium and transformed spontaneously after long passages [34]. Nevertheless, ID8 cannot generate HGSOC phenotypes due to not carrying the characteristic driver mutations of HGSOC (e.g., *Trp53* and *Rb*, etc.). Quartuccio et al. studied spontaneous transformed murine oviductal epithelial (MOE)^hi^ cells after long passages of MOE^low^ and could generate tumors after long-term allograft. However, the tumor exhibited a sarcoma phenotype [35]. To our knowledge, FE25L is the first HGSOC cell line that spontaneously transformed from human fimbria secretory cells with the characteristic *p53* and *Rb/CCNE1* defect and CNV phenotypes.

Aggressive cancer cells often exhibit a high proliferation rate [36]. High-passage cancer cells also exhibit a shorter doubling time [37]. Our study was also consistent with previous studies in that FE25L showed a higher proliferation rate than that of FE25 cells.

Cancer chromosome instability (CIN) causes cell chromosome number and structural changes, resulting in tumor heterogeneity [38]. In most solid tumors, CIN with aneuploidy is associated with drug resistance and poor prognosis [39,40]. In our study, both FE25 and FE25L showed these features of CIN with DNA polyploidy and aneuploidy. Both cells showed loss of chromosome 19 commonly found in ovarian cancer [41], and showed loss of chromosome 9 (which is typically associated with copy number variation) [39], 14, 16, and 18.

Compared to FE25, FE25L cells showed more severe CIN with progressive CNV. HGSOC carries a broad spectrum of diverse alterations in CNV [42]. CNV deletion of *Tp53* causes further suppression or misdirection of p53 [43]. Among the FE25/FE25L-altered genes, *MYC* (the stem-cell transcription factor) was the most upregulated gene (42% with a 4N copy number, 37% with 3N) in TCGA. Additionally, Her2 (encoded by *ERBB2*) overexpression can be noted but is unrelated to CNV amplification [44]. Comparative genomic hybridization revealed that *CCNE1* was significantly amplified [45]. Nevertheless, few additional single-gene drivers have been explored in TCGA [16,46]. Our study also found more CNV amplifications of *ERBB2*, *TP53*, and *CCNE1* in FE25L than in FE25 cells.

*CASP1* is one of the FE25L-to-FE25 copy number-altered genes which was confirmed at RNA level. Caspase 1 (*casp1*) is an apoptosis-related protein [47]. A previous study found that caspase 1 was abundant in the normal ovarian surface epithelium but was reduced in ovarian cancer cell lines [47]. Therefore, the downregulation of caspase 1 may be related to increased resistance to apoptosis in ovarian cancer cells. Our study showed *casp1* was downregulated in FE25L cells (CNV-low), which may be related to the characteristics of advanced ovarian cancer.

Hypoxia that influences invasion and adhesion functions can promote ovarian cancer proliferation [48]. Patients with ovarian cancer with a high EMT index were associated with worse overall survival than patients with a low EMT index [20]. EMT is an important process in cancer invasion and metastasis and involves more than 30 gene expressions [49]. Cell morphology and cell-cell adhesion also affect invasion and migration [50,51]. NF-κB is also associated with decreased survival rate in ovarian cancer [23]. In our RNA-seq data, the three most significantly enriched gene sets in upregulated genes included hypoxia, EMT, and the NF-κB pathway in FE25L compared to FE25. qRT-PCR also confirmed that FE25L cells had increased expression of EMT-related genes, including v*imentin, fibronectin*, and *Twist*.

Gene expression of cytokines also indicates the malignant behavior of ovarian cancer cells. Elevated levels of cytokines in ascites are associated with patient survival [52]. We previously showed that IL-6 is increased in ovarian stromal cells and enhances the aggressive behavior of ovarian cancer [53]. Insulin-like growth factor (IGF) is also important for ovarian cancer initiation [12]. In this study, we showed increased expression of *IL-6*, *IL-8*, *IGF-2*, and *VEGFA* in FE25L cells compared to that in FE25 cells. These cytokines also correlate with the aggressive tumorigenic behavior of the cells.

In ovarian cancer, the Wnt/β-catenin pathway is one of the significant signaling pathways. A previous study showed that increased β-catenin expression reduces Dicer expression to promote ovarian cancer metastasis [54]. Conversely, inhibition of the β-catenin pathway can inhibit ovarian cancer metastasis [55]. Therefore, β-catenin is believed to be involved in ovarian cancer metastasis. In our study, after long-term in vitro passage of FE25 cells, FE25L cells exhibited increased β-catenin signaling. Therefore, our results show that FE25L exhibited an increased metastatic capability compared to FE25 cells.

Cancer metastasis is represented in the late stage and is related to a poor prognosis [56]. Metastasis is a process that involves cancer cell invasion and migration. The colony formation ability of cancer cells revealed their invasiveness. In our study, FE25L showed increased migration, invasion, and colony formation capabilities.

Following debulking surgery, adjuvant chemotherapy with carboplatin and paclitaxel is administered. However, despite the high response rate to the first chemotherapy, the disease often recurs with chemoresistance [57]. Therefore, drug resistance in cancer cell lines is also a marker of aggressive cancer behavior [58]. However, in our study, the IC_50_ values of the three chemotherapeutic drugs were lower in FE25L than in FE25 cells. Low drug resistance may be due to the CNV status of FE25L.

The Cancer Genome Atlas can systematically compare genomic features of various cell lines and tumors. The previous study showed significant differences in molecular profiles between commonly used ovarian cancer cell lines and HGSOC tumor samples [27]. The results revealed that a gap between cell lines and tumors could be bridged by genomically choosing cell line models for HGSOC. Cell lines such as TYKNU and COV362 are top-ranked HGSOC-like cell lines other than commonly used A2780 cells [27]. Unsupervised hierarchical clustering analysis of RNA expression showed FE25L was more clustered with cell lines such as TYNKU and CAOV3 (likely HGSOC), which are characterized as better cell lines for studying human HGSOC.

The main limitation of this study was that most of the experiments were more than three technical replicates but not different cell batch replicates. Nevertheless, the FE25 cells has been extensively studied in our previous research in different cell batches with different passage numbers [28,29,31,32]. They showed consistent results. Therefore, the current results could be trusted.

## 4. Materials and Methods

### 4.1. Cell Culture

FE25 and FE25L (high passage) cells were obtained from Dr. Chu TY’s lab and were cultured in MCDB105 and M199 medium (Sigma, St. Louis, MO, USA) supplemented with 10% fetal bovine serum (FBS), 100 IU/mL penicillin, and 100 μg/mL streptomycin. 5 × 10^5^ cells were grown in 75 cm^2^ culture flasks in an incubator maintained at 37 °C and 5% CO_2_. The culture medium was replaced every 2 days. Until reaching 75% of confluence, they were passaged at 1:3 ratio with trypsin (Sigma) treatment.

### 4.2. Cell Proliferation Assay

The cell proliferation kit (XTT (2,3-Bis-(2-Methoxy-4-Nitro-5-Sulfophenyl), Biological Industries Ltd., Kibbutz Beit Haemek, Israel) assay was used to measure cell proliferation according to the manufacturer’s instructions. After 0, 3, 5, and 7 days of incubation, proliferation with XTT was tested. Absorbance was detected using a microplate reader at an optical density of 450 nm (Bio-Rad Model 3550, Hercules, CA, USA). The optical density values at each time point were used to construct proliferation curves of the tested cell lines.

### 4.3. Cell Cycle Analysis

FE25 and FE25L cells were detached using 0.1% trypsin and resuspended in phosphate-buffered saline (PBS, Sigma) to a final concentration of 5 × 10^6^ cells/mL. 1 mL of the cell suspension was washed twice with ice-cold PBS. The cells were fixed with 70% ethanol on ice for one hour and washed twice with ice-cold PBS. Cells were then incubated with 1 mL propidium iodide (20 μg/mL, BD Biosciences, Franklin Lakes, NJ, USA), Triton X-100 (0.1%, Invitrogen), and RNase A (0.2 mg/mL, Invitrogen) in the dark for 30 min at 0 °C. A cell cycle distribution was studied by acquiring 10,000 events per specimen using a flow cytometer (BD Biosciences, San Jose, CA, USA). The percentage of cells in the G0/G1, S, and G2/M phases was calculated by Flowjo software (version 10.8, BD Bioscience).

### 4.4. Chromosome Evaluation

Aneuploidy is a characteristic feature of cancer cells [59]. Aneuploidy was studied in FE25 (passage 31) and FE25L (passage 139) cells at the cytogenetics core laboratory at Hualien Tzu Chi Hospital, Hualien, Taiwan. Cells were cultured in the medium until they reached an adequate amount, and 0.25% colchicine (Sigma) was added to arrest cells at metaphase. The hypotonic solution was then added to induce cell bursting. Giemsa (1:20, Sigma) staining chromosomes were reviewed by a cytogeneticist. After counting 50 metaphases, the chromosome number distribution was acquired and reported according to the 2016 International System for Human Cytogenetic Nomenclature.

### 4.5. Colony Formation Assay

We plated 200 cells in a 2 mL culture medium in one well of a 6-well plate. After 14 days of culture, the cells were fixed with 70% ethanol and stained with 0.8 mM crystal violet (Sigma-Aldrich), and the number of colonies was measured by ImageJ (NIH, Bethesda, MA, USA). The experiments were performed in triplicate

### 4.6. Migration and Invasion Assay

We placed 5 × 10^4^ cells in 200 μL medium into the upper chamber of a 24-well trans-well Boyden chamber (8 μm pore size; Costar, Corning Inc., Corning, NY, USA), and allowed the cells to migrate to the lower layer, where there were no cells and which only had 500 μL culture medium. After culturing for 48 h, the cells were stained with DAPI (Sigma-Aldrich) and counted using an immunofluorescence microscope. The experiments were performed in triplicate.

### 4.7. Drug Sensitivity Analysis

Paclitaxel and doxorubicin are common chemotherapeutic drugs used to treat ovarian cancer [60]. The effects of paclitaxel (Formoxol, Yung Shin Pharm. Ind., Co. Ltd., Taichung, Taiwan), doxorubicin (Adriblastina, Pfizer, Kent, NJ, USA), and carboplatin (SINPHAR Pharmaceutical Co., Ltd., Yilan, Taiwan) on FE25 and FE25L cells were studied. Cells were seeded at a density of 2500–3000 cells/well in a 96-well plate, cultured overnight, and treated with chemotherapeutic drugs (taxol: 41.0 and 0.39 nM, doxorubicin: 3.1 and 1.4 μM, carboplatin: 26.4 and 2.0 μM for FE25 and FE25L, respectively) for 48 h.

XTT solution was used for calculating cell proliferation according to the manufacturer’s instructions. After incubation for 2–5 h at 37 °C, the absorbance of the wells was determined using a spectrophotometer (DYNEX MRX II ELISA reader, Bustehrad, Czech) at a wavelength of 450 nm and a reference wavelength of 650 nm.

The IC_50_ values of the three drugs for these two cell lines were then obtained using a non-linear regression model on GraphPad Prism (version 9.0, GraphPad Software, San Diego, CA, USA).

### 4.8. Gene Expression Analysis by qRT-PCR

#### Extraction of Total RNA from Cells

RNeasy Mini kit (QIAGEN, Hilden, Germany) was used to isolate RNA from 5 × 10^5^ FE25 or FE25L cells plated in a 10-cm culture dish. Briefly, after 24 h of culture, the medium was removed and the cells were washed twice with 1 × PBS, detached using 0.05% trypsin, and pelleted. The cells were then lysed in 700 μL RLT lysis buffer by aspirating them several times with a 1 mL micropipette until the cells were completely lysed and the solution became transparent. Reagents were then added to extract RNA according to the manufacturer’s instructions.

### 4.9. Preparation of cDNA

One μg of total RNA was used for cDNA synthesis using Reverse-iT^TM^ 1st strand synthesis kit (ABgene, Portsmouth, NH, USA) using the manufacturer’s instructions. Briefly, 1μg total RNA and 1 μL anchored oligo dT primers were mixed with DEPC-treated (diethylpyrocarbonate) water to a total volume of 12 μL, incubated at 70 °C for 5 min, placed on ice. Then 8 μL reaction mixture (4 μL 5× First-strand synthesis buffer, 2 μL dNTP mix 5 mM each, 1 μL 100 mM DTT, Reverse-iT^TM^ RTase blend 1 μL) was added followed by incubation at 47 °C for 50 min, after which the reaction was terminated at 75 °C for 10 min, followed by storage at −20 °C until use.

#### Quantitative Polymerase Chain Reaction

We used the ABI Step One Plus system (Applied Biosystems, Waltham, MA, USA) and FastStart Universal SYBR Green Master (ROX, Basel, Switzerland) gene expression analysis reagents for quantification of gene expression. mRNA expression levels were normalized to the mean Ct of the GAPDH, which was used as an endogenous control. The primer sequences are listed in Table 5.

### 4.10. RNA Sequencing and Gene Set Enrichment Analysis (GSEA)

Gene expression profiling was conducted using high-throughput RNA sequencing (Illumina NovaSeq 6000 platform) with 28,264 human genome probes. Upregulated and downregulated genes in FE25L compared to FE25 were identified and subjected to gene ontology analysis using GSEA [19]. Online GSEA (https://www.gsea-msigdb.org/gsea/index.jsp, assessed on 19 May 2022) was used to identify gene function sets with a significant false detection rate (FDR) q-value of <0.05.

### 4.11. Cross-Platform Comparability

Cluster analysis of FE25, FE25L, and ovarian cancer cell lines, including A2780, CP70, A1847, KURAMOCHI, etc., was performed. Human OneArray version HOA 6.1 (Phalanx Biotech Group, Hsinchu, Taiwan) platform containing 28,264 human genome probes was used. Additionally, public domain data of ovarian cancer cell lines (Cancer Cell Line Encyclopedia; CCLE RNAseq gene expression data for 1019 cell lines) and the Affymetrix Human Genome U133 Plus 2.0 Array (Platform GPL 15308) were used to cluster FE25 and FE25L with the analyzed ovarian cancer cell lines. The previous study grouped current ovarian cancer cell lines into group 1 (likely HGSOC), group 2 (probably HGSOC), and group 3 (unlikely HGSOC) [27]. We used the group 1 genetic feature to identify the cell lines that resemble HGSOC cells. The top 500 diverse genes of group 1 were selected to differentiate the difference between the cell lines.

### 4.12. Library Preparation for the Whole Exon Sequencing

The DNA library was prepared using 1 μg of genomic DNA as input material. An Agilent SureSelect Human All Exon kit (Agilent Technologies, Santa Clara, CA, USA) was used to generate sequencing libraries. A hydrodynamic shearing system (Covaris, Woburn, MA, USA) was used to generate 160–280 bp DNA fragments. The exonuclease/polymerase activities were used to convert the remaining overhangs into blunt ends. The 3′ ends of the DNA fragment were adenylated and adapter oligonucleotides were ligated. Polymerase chain reaction (PCR) was used to enrich the DNA fragments and ligate adapter molecules at both ends. After the PCR, the biotin-labeled probe in the liquid phase was hybridized into the library. The exons of the genes were captured using streptomycin-coated magnetic beads. PCR was used to enrich the captured libraries for hybridization. An AMPure XP system (Beckman Coulter, Beverly, MA, USA) was used to purify the final products. An Agilent high-sensitivity DNA assay on the Agilent Bioanalyzer 2100 system was used to quantify the products.

### 4.13. Copy Number Detection and Visualization

To detect copy number variation (CNV) in FE25 and FE25L, a software toolkit CNVkit (v0.9.8) was used to detect and visualize CNVs from whole-exome sequencing data [61]. Briefly, to determine the genome-wide copy number, the read depths of the on-target sequencing reads were compared and provided comparable copy number estimates in the target regions. The normalized copy ratio was visualized using a scatter plot.

### 4.14. Western Blot

Sodium dodecyl sulfate polyacrylamide gel electrophoresis (SDS-PAGE) was performed in 1× running buffer (0.025 M Tris, 0.192 M glycine, 0.1% SDS) as the electrolytic buffer. The extracted protein (50 μg), and 6× protein-loading dye were boiled at 95 °C for 10 min, and 18 μL/well was loaded into the gel. The gel was first run at a voltage of 80 V and after the sample passed through the stacking gel, the voltage was increased to 120 V. After electrophoresis was completed, a transfer buffer was used to transfer the protein separated by SDS-PAGE colloid to the transfer membrane. The transfer membrane was washed with 1× PBST (PBS Tween-20 buffer, 0.13 M NaCl, 0.01 M NaH2PO4, 0.05% (*v*/*v*) Tween 20) for 10 min, followed by incubation in the blocking buffer for 1 h. The membranes were then incubated with the primary antibody diluted in the blocking buffer for 1 h. The samples were washed three times with PBST for 10 min. Membranes were then incubated with HRP-linked secondary antibody diluted in blocking buffer for 1 h, and washed three times with PBST for 10 min. The protein bands were visualized by incubating in HRP substrate in the dark for 1 min. The samples were clamped in a plastic bag and detected using a luminometer (FUJIFILM LAS-3000). The antibodies used were β-catenin (1:1000, Sigma-Aldrich). β-actin (1:10,000, Sigma-Aldrich) and histone-H3 (1:10,000, Sigma-Aldrich) were used as internal controls.

For protein quantification, the data were normalized to total protein loading (actin or histone-H3), and relative quantification was compared with that of the original cells for fold-change analysis. ImageJ software (National Institutes of Health, Bethesda, MA, USA) was used to perform quantification according to the methods described in the previous study [62].

The ReadyPrep protein extraction kit (cytoplasmic/nuclear, Bio-Rad) was used to isolate proteins from the nucleus and cytoplasm following the manufacturer’s instructions. Western blotting was performed to analyze the resulting cytoplasmic/nuclear proteins.

### 4.15. Xenograft Tumorigenesis

NSG immunodeficient mice [NOD/Shi-*scid*/IL-2Rγ^null^] were purchased from Jackson Laboratory and raised at the Tzu Chi University Laboratory Animal Center (Hualien, Taiwan). The Laboratory Animal Care and Use Committee of Hualien Tzu Chi Hospital approved the experimental protocol.

Mice were divided into two groups: FE25 and FE25L (*n* = 3 each). A total of 1 × 10^6^ tested cells were transplanted into the subcutaneous region of each mouse’s back. We observed by palpation and measured the tumor growth in each group of mice. Tumor volume was calculated as (width)^2^ × length/2 (mm^3^). Three to six months later, when the tumor volume of the mouse was greater than 500 mm^3^, the mice were sacrificed, and the tumor was removed and stored in neutral formalin.

The percentage of tumorigenesis and xenograft proliferation rate of the test cells was calculated and compared between the two groups.

### 4.16. Immunohistochemistry (IHC)

Hematoxylin and eosin staining and immunostaining were used to evaluate the histology of the tumors formed by the tested cells. First, xylene was used to remove the formalin, followed by alcohol stratification and rehydration. The antigen unmasking solution (Dako, Agilent, Santa Clara, CA, USA) and 3% hydrogen peroxide in distilled water were used to block the intrinsic peroxidase activity. Immunostaining was performed with antibodies against PAX8 (1:500, Dako) and WT-1 (1:500, Dako), CK7 (1:500, Dako) according to the manufacturer’s instructions (Vector). The specimens were then placed in a refrigerator at 4 °C overnight and treated with anti-rabbit horseradish peroxidase-labeled polymer (Dako) at room temperature for 30 min. All immunostained slides were stained with hematoxylin. WT-1 positive cells were calculated by the positive staining cell numbers among 50 cells counted in each of the three fields.

### 4.17. Statistical Analysis

This study (proliferation, gene expression, colony number, migration and invasion cell numbers, and tumor proliferation rate) used the non-parametric test (Mann–Whitney U test) for two groups to analyze whether the experimental results were statistically significant. Fisher’s exact test was used to calculate the percentage of tumorigenesis. Statistical significance was set at 0.05. The results are presented as mean ± standard deviation (SD).

## 5. Conclusions

We demonstrated that FE25L cells showed more aggressive behavior in cancer than FE25. Genome instability, proliferation rate, migration, invasion, colony formation, and tumor formation capabilities were higher in the FE25L group than in the low passage FE25 cells. More HGSOC gene signatures and CNV were observed in the FE25L group. Therefore, FE25L might be a more suitable model for ovarian cancer research. Future studies should aim at comprehensively analyzing the differences in gene expression between FE25L and FE25 cells.

## Figures and Tables

**Figure 1 ijms-23-13843-f001:**
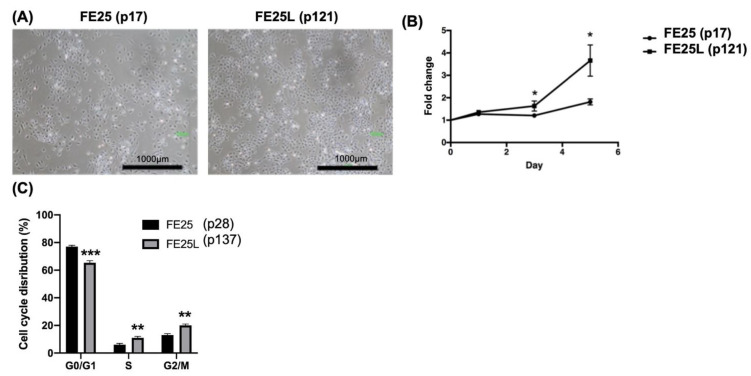
The characteristics of FE25 (p17) and FE25L (p121) cells. (**A**) The morphology of FE25 and FE25L cells. Scale bar = 1000 µm. (**B**) Proliferation curve of the FE25 and FE25L cells over 5 days (*n* = 3 in each group). The error bars represent standard deviation (SD). Data were expressed as fold change in cell numbers compared to day 0 (value = 1). * *p* < 0.05. (**C**) Distribution curves showing quantitative proportion of the cells in each cell cycle in at least 10,000 cells of each sample (*n* = 3). The error bar represents the mean ± SD. All the experiments were performed in triplicates. ** *p* < 0.01, *** *p* < 0.001.

**Figure 2 ijms-23-13843-f002:**
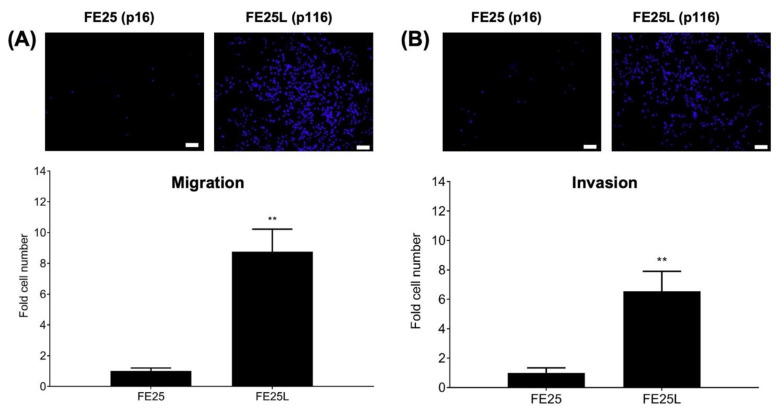
Migration and invasion assay in FE25 (p16) and FE25L (p116) cell lines. (**A**) Trans-well migration assay. Culture medium 500 μL was added to lower chamber. After cells migrated for 48 h, cells on the membranes were stained with DAPI. Blue immunofluorescence-stained migrated cells were pictured by a microscope (upper panel). Then migrated cells were quantified and compared between the two cell lines (total migrated cells on the membranes) (*n* = 3 in each group). (**B**) Trans-well invasion assay. Upper chamber was coated with Matrigel. The other steps of the experiment were the same as described above (*n* = 3 in each group). ** *p* < 0.01. Scale bar = 100 μm. The upper panel shows a representative image of the experiment.

**Figure 3 ijms-23-13843-f003:**
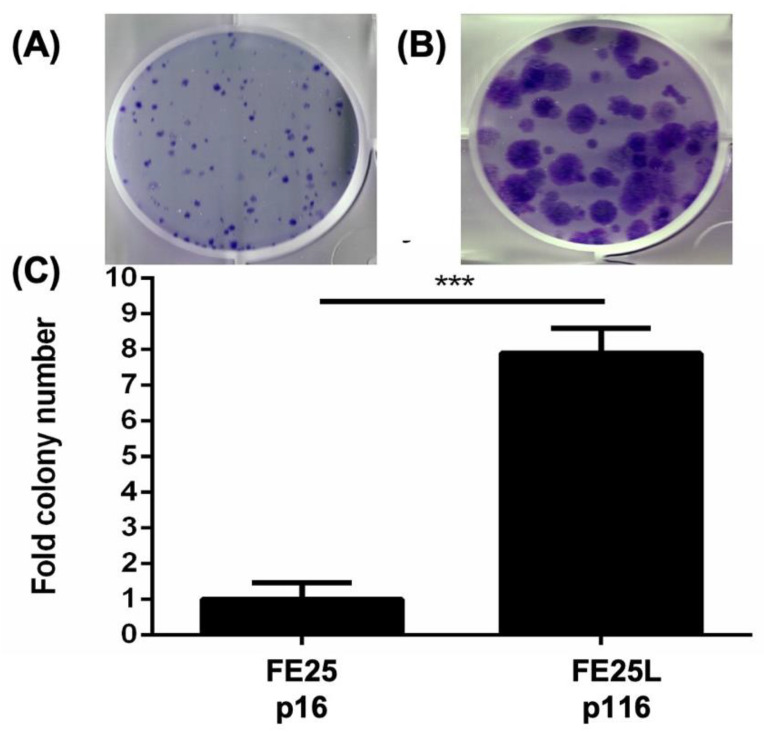
Colony formation assay for FE25 (p16) and FE25L (p116) cells. 200 cells were plated in 2 mL of culture medium in one well of a six-well plate. After 14 days, the colonies were stained with crystal violet and counted. (**A**,**B**) Representative colony growth of FE25 (**A**) and FE25L **(B)** cells. (**C**) Quantification of the colony numbers in three replicates. *** *p* < 0.001.

**Figure 4 ijms-23-13843-f004:**
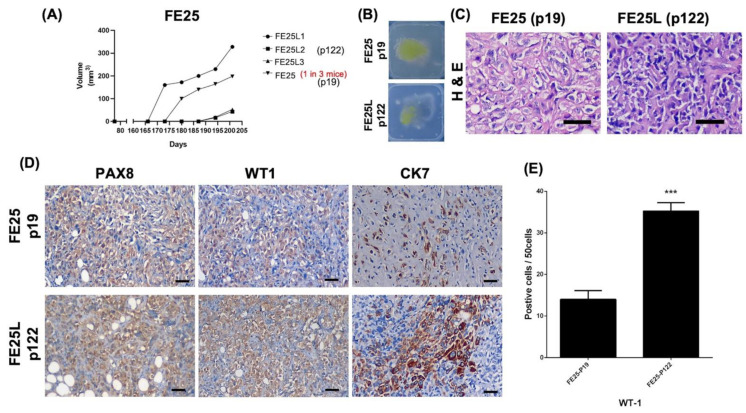
FE25L cells (p122) generate more tumors than FE25 cells (p19) in a xenograft model. NSG female mice (4–5 weeks of age) were subcutaneously injected with 1 × 10^6^ FE25 or FE25L cells (*n* = 3 in each group). (**A**) Tumor growth curve. Tumors appeared after 165 and 176 days in FE25L and FE25, respectively. All three mice injected with FE25L resulted in tumors (3/3) whereas only 1 mouse showed tumor in the FE25 injected group (1/3). (**B**) Gross xenograft tumor morphology of FE25 and FE25L in wax blocks. (**C**) H & E staining of the xenografts obtained from FE25 and FE25L injected mice. Scale bar = 100 μm. (**D**) Immunohistochemistry for fallopian tube epithelium marker, PAX8, WT1, and CK7 for FE25 and FE25L tumors. Scale bar = 100 μm. (**E**) The comparison of WT1 expression between FE25 and FE25L (*n* = 3 in each group). *** *p* < 0.001.

**Figure 5 ijms-23-13843-f005:**
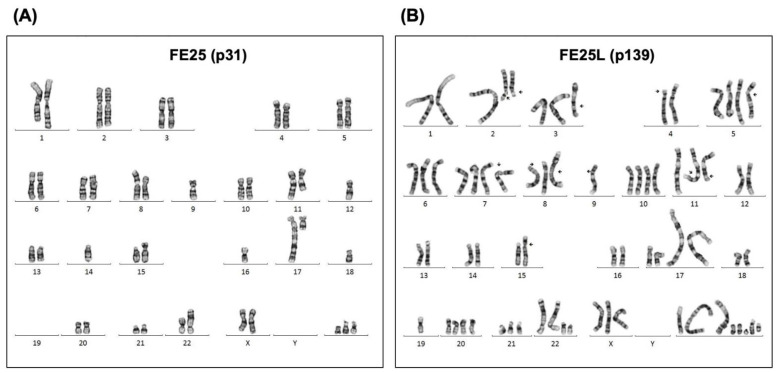
G-band karyotyping for FE25 (p31) and FE25 L (p139) cells. Representative karyotyping chromosome images of (**A**) FE25 and (**B**) FE25L cells. Numbers under each chromosome were chromosome numbers.

**Figure 6 ijms-23-13843-f006:**
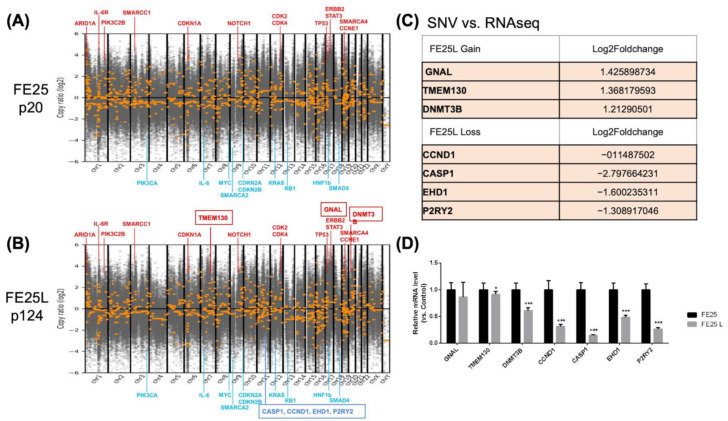
DNA copy number of FE25 (p20) and FE25L (p124) cells revealed by whole exome sequencing. (**A**,**B**) The position of the chromosome is listed in *x*-axis. *y*-axis showed the gain or loss of copy number in log_2_. Red color line with gene name indicates amplified loci and water blue color line with gene name indicates deleted loci of the two cell lines as compared to the normal genome. (**C**) Table shows genes with somatic varied copy number (SNV) upon FE25 to FE25L progression, which also showed a compatible mRNA alteration as revealed by RNA sequencing. (**D**) The genes listed in Table (**C**) were further confirmed by qRT-PCR. *** *p* < 0.001. * *p* < 0.05.

**Figure 7 ijms-23-13843-f007:**
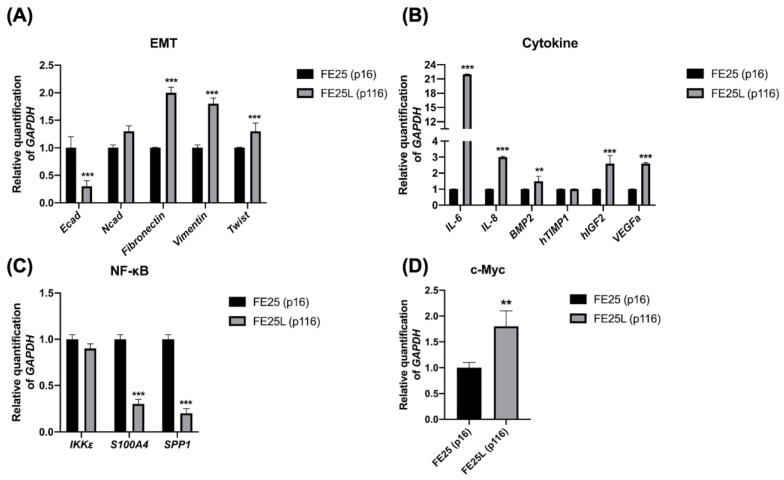
Quantitative RT-PCR validation of gene expression of FE25 (p16) and FE25L (p116) cells. qRT-PCR analysis for (**A**) epithelial-mesenchymal transition (EMT), (**B**) cytokines, (**C**) NF-κB signaling, and (**D**) c-myc is presented for FE25 and FE25L cells (*n* = 3 in each group). Error bars represent SD. ** *p* < 0.01, *** *p* < 0.001. RQ: relative quantification to GAPDH.

**Figure 8 ijms-23-13843-f008:**
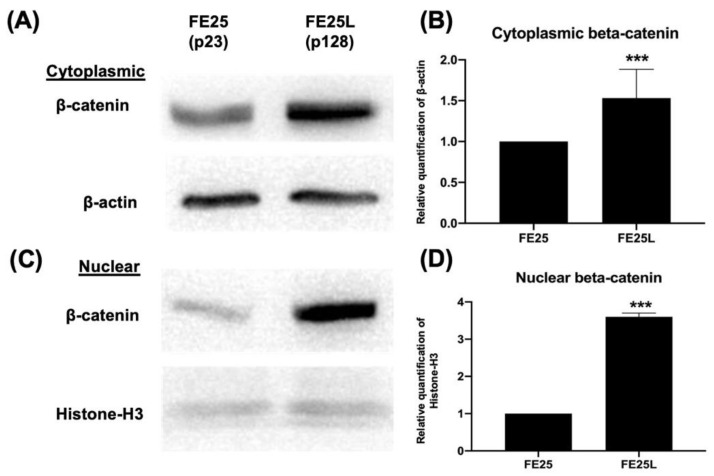
Beta-catenin gene and protein expression in FE25 (p23) and FE25L (p128) cells. (**A**) Western blot analysis of beta-catenin protein expression in the cytoplasm of FE25 and FE25L cells. (**B**) Quantification of beta-catenin expression in (**A**) (*n* = 3 in each group). (**C**) Western blot analysis of beta-catenin expression in the nuclei of FE25 and FE25L cells. (**D**) Quantification of beta-catenin expression in (**C**) (*n* = 3 in each group). *** *p* < 0.001.

**Figure 9 ijms-23-13843-f009:**
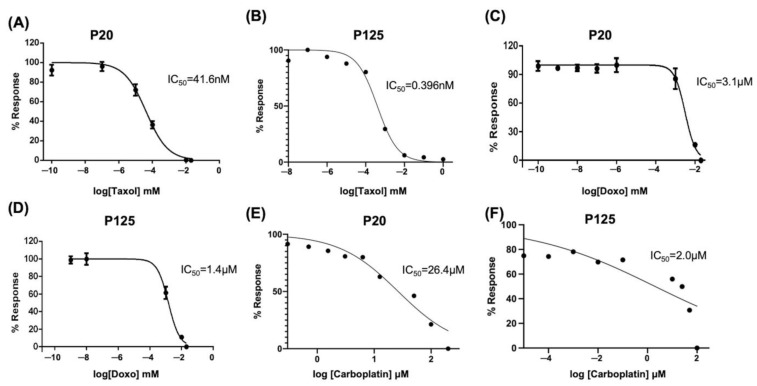
Chemosensitivity analysis of FE25 (p20) and FE25L (p125) cell lines. The IC_50_ curve of taxol at passage 20 (**A**) and passage 125 (**B**); doxorubicin at passage 20 (**C**) and passage 125 (**D**); carboplatin at passage 20 (**E**) and passage 125 (**F**) FE25 cells. All experiments were repeated three times. Data represents mean ± standard deviation for three independent experiments.

**Figure 10 ijms-23-13843-f010:**
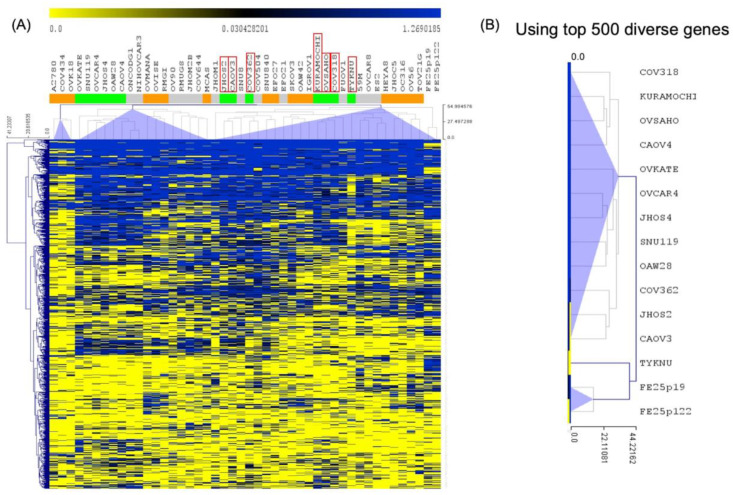
Unsupervised hierarchical clustering analysis of RNA expression. (**A**) Clustered image map of FE25 (p19) and FE25L (p122) by analyzing mRNA expression levels. (**B**) The most correlated group 1 HGSOC lines were JHOS2, CAOV3, COV362, and TYKNU using the top 500 diverse genes of group 1 HGSOC (likely HGSOC) cell lines.

**Table 1 ijms-23-13843-t001:** CNV gain sites and overlap genes of FE25L (p124) compared to FE25 (p20) cells.

Gene Set Name [# Genes (K)]	Description	# Genes in Overlap (k)	*p*-Value	FDR q-Value
NIKOLSKY_BREAST_ CANCER_7Q21_Q22_AMPLIC	Genes within amplicon 7q21–q22 identified in a copy number alterations study of 191 breast tumor samples.	*ARPC1A*, *PDAP1*, *ATP5MF*, *BUD31*, *ARPC1B*, *BAIAP2L1*, *SMURF1*, *ZKSCAN5*, *TRRAP*, *PTCD1*, *FAM200A*, *CPSF4*, *ZNF789*, *ZNF394*, *NPTX2*, *TMEM130*, *LMTK2*, *BRI3*, *TECPR1*, *BHLHA15*	1.97 × 10^−37^	6.37 × 10^−33^
chr18p11 <201>	Ensembl 103 genes in cytogenetic band chr18p11	*RAB31*, *RALBP1*, *VAPA*, *NDUFV2*, *PPP4R1*, *TWSG1*, *PTPRM*, *MTCL1*, *ANKRD12*, *GNAL*, *RAB12*, *NAPG*, *LRRC30*, *APCDD1*, *NDUFV2-AS1*, *PIEZO2*, *TXNDC2*, *MIR7153*, *MIR6788*, *SLC35G4*	3.62 × 10^−28^	5.85 × 10^−24^
chr7q22 <213>	Ensembl 103 genes in cytogenetic band chr7q22	*ARPC1A*, *PDAP1*, *ATP5MF*, *BUD31*, *ARPC1B*, *BAIAP2L1*, *SMURF1*, *ZKSCAN5*, *TRRAP*, *PTCD1*, *FAM200A*, *CPSF4*, *ZNF789*, *ZNF394*, *NPTX2*, *TMEM130*, *ATP5MF-PTCD1*, *KPNA7*, *MYH16*, *MIR3609*	1.2 × 10^−27^	1.29 × 10^−23^
chr2p25 <117>	Ensembl 103 genes in cytogenetic band chr2p25	*ACP1*, *FAM110C*, *SNTG2*, *TPO*, *SH3YL1*, *TMEM18*, *ALKAL2*	5.92 × 10^−9^	4.78 × 10^−5^
chr20q11 <237>	Ensembl 103 genes in cytogenetic band chr20q11	*DNMT3B*, *MAPRE1*, *COMMD7*, *AL034550.2*, *NOL4L*, *EFCAB8*, *C20orf203*, *NOL4L-DT*	4.13 × 10^−8^	2.67 × 10^−4^
DIAZ_CHRONIC_ MEYLOGENOUS_LEUKEMIA_UP <1398>	Genes up-regulated in CD34+ [GeneID = 947] cells isolated from bone marrow of CML (chronic myelogenous leukemia) patients, compared to those from normal donors. (chr7)	*ARPC1A*, *PDAP1*, *ATP5MF*, *RAB31*, *RALBP1*, *VAPA*, *NDUFV2*, *PPP4R1*, *TWSG1*, *FAM110C*, *DNMT3B*, *URI1*, *ZMYND11*	7.21 × 10^−6^	3.88 × 10^−2^

**Table 2 ijms-23-13843-t002:** CNV loss sites and overlap genes of FE25L (p124) compared to FE25 (p20) cells (all at chr11).

Gene Set Name [# Genes (K)]	Description	# Genes in Overlap (k)	*p*-Value	FDR q-Value
HALLMARK_ COMPLEMENT <200>	Genes encoding components of the complement system, which is part of the innate immune system.	*SERPING1*, *MMP8*, *S100A13*, *FDX1*, *CTSC*, *CASP1*, *CASP4*, *S100A9*, *EHD1*, *APOA4*, *MMP12*, *MMP13*, *PRCP*, *S100A12*, *CASP5*, *RCE1*	1.99 × 10^−4^	9.95 × 10^−3^
HALLMARK_ COAGULATION <138>	Genes encoding components of blood coagulation system; also up-regulated in platelets.	*SERPING1*, *MMP8*, *S100A13*, *PRSS23*, *S100A1*, *MMP1*, *MMP3*, *MMP10*, *APOA1*, *CAPN5*, *APOC3*, *MMP7*	5.75 × 10^−4^	1.44 × 10^−2^
HALLMARK_OXIDATIVE_PHOSPHORYLATION <200>	Genes encoding proteins involved in oxidative phosphorylation.	*FDX1*, *COX8A*, *DLAT*, *CPT1A*, *SDHD*, *TCIRG1*, *ATP5MG*, *TIMM10*, *TIMM8B*, *ACAT1*, *NDUFC2*, *NDUFV1*, *NDUFS8*, *MRPL11*	1.77 × 10^−3^	2.95 × 10^−2^
HALLMARK_ESTROGEN_ RESPONSE_EARLY <200>	Genes defining early response to estrogen.	*PRSS23*, *DHCR7*, *CCND1*, *PLAAT3*, *ALDH3B1*, *PGR*, *ENDOD1*, *P2RY2*, *TSKU*, *RHOD*, *NADSYN1*, *SYT12*, *GAB2*	4.77 × 10^−3^	4.77 × 10^−2^
HALLMARK_MTORC1_ SIGNALING <200>	Genes up-regulated through activation of mTORC1 complex.	*CTSC*, *DHCR7*, *SC5D*, *FADS2*, *TM7SF2*, *SERPINH1*, *RPA1*, *HMBS*, *STIP1*, *FADS1*, *SLC37A4*, *FKBP2*, *SYTL2*	4.77 × 10^−3^	4.77 × 10^−2^

FDR: false discovery rate.

**Table 3 ijms-23-13843-t003:** GSEA based on the hallmark gene sets for upregulated genes of FE25L (p122) compared to FE25 (p19) cells.

Gene Set Name [# Genes (K)]	Description	Genes in Overlap	FDR q-Value
HALLMARK_HYPOXIA	Genes up-regulated in response to low oxygen levels (hypoxia).	*VEGFA*, *CDKN1A*, *BTG1*, *NDRG1*, *DDIT3*, *DDIT4*, *ADORA2B*, *NR3C1*, *MXI1*, *ADM*, *SRPX*, *HOXB9*	5.42 × 10^−12^
HALLMARK_EPITHELIAL_ MESENCHYMAL_TRANSI NSITION	Genes defining epithelial-mesenchymal transition, as in wound healing, fibrosis, and metastasis.	*VEGFA*, *COL6A2*, *COL6A3*, *ADAM12*, *TFPI2*, *MXRA5*, *LAMA1*, *SERPINE2*, *SFRP1*	3.14 × 10^−8^
HALLMARK_TNFA_SIGNALING_VIA_NFKB	Genes regulated by NF-κB in response to TNF	*VEGFA*, *CDKN1A*, *BTG1*, *PHLDA1*, *ETS2*, *ABCA1*, *DUSP5*, *EGR1*, *MSC*	3.14 × 10^−8^
HALLMARK_P53_PATHWAY	Genes involved in p53 pathways and networks.	*CDKN1A*, *BTG1*, *NDRG1*, *DDIT3*, *DDIT4*, *TRIB3*, *F2R*, *UPP1*	4.86 × 10^−7^
HALLMARK_IL2_STAT5_SIGNALING	Genes up-regulated by STAT5 in response to IL2 stimulation.	*NDRG1*, *PHLDA1*, *NRP1*, *LRIG1*, *F2RL2*, *PLAGL1*, *SCN9A*	6.78 × 10^−6^
HALLMARK_PI3K_AKT_MTOR_ SIGNALING	Genes up-regulated by activation of the PI3K/AKT/mTOR pathway.	*CDKN1A*, *DDIT3*, *TRIB3*, *CALR*, *CAMK4*	5.5 × 10^−5^
HALLMARK_MTORC1_SIGNALING	Genes up-regulated through activation of mTORC1 complex.	*CDKN1A*, *DDIT3*, *DDIT4*, *TRIB3*, *CALR*, *NIBAN1*	7.8 × 10^−5^
HALLMARK_ANGIOGENESIS	Genes up-regulated during formation of blood vessels (angiogenesis).	*VEGFA*, *NRP1*, *FGFR1*	5.47 × 10^−4^
HALLMARK_HEDGEHOG_SIGNALING	Genes up-regulated by activation of hedgehog signaling.	*VEGFA*, *ETS2*, *NRP1*	5.47 × 10^−4^
HALLMARK_UNFOLDED_PROTEIN_ RESPONSE	Genes up-regulated during unfolded protein response, a cellular stress response related to the endoplasmic reticulum.	*VEGFA*, *DDIT4*, *CALR*, *HYOU1*	9.18 × 10^−4^
HALLMARK_UV_RESPONSE_DN	Genes down-regulated in response to ultraviolet (UV) radiation.	*ADORA2B*, *NR3C1*, *NRP1*, *BCL2L11*	2.1 × 10^−3^
HALLMARK_APOPTOSIS	Genes mediating programmed cell death (apoptosis) by activation of caspases.	*CDKN1A*, *DDIT3*, *F2R*, *BCL2L11*	2.93 × 10^−3^
HALLMARK_CHOLESTEROL_ HOMEOSTASIS	Genes involved in cholesterol homeostasis.	*TRIB3*, *NIBAN1*, *CBS*	3.21 × 10^−3^
HALLMARK_GLYCOLYSIS	Genes encoding proteins involved in glycolysis and gluconeogenesis.	*VEGFA*, *DDIT4*, *ADORA2B*, *MXI1*	4.89 × 10^−3^
HALLMARK_INFLAMMATORY_ RESPONSE	Genes defining inflammatory response.	*CDKN1A*, *ADORA2B*, *ADM*	4.89 × 10^−3^
HALLMARK_MYOGENESIS	Genes involved in development of skeletal muscle (myogenesis).	*CDKN1A*, *COL6A2*, *COL6A3*, *ADAM12*	4.89 × 10^−3^
HALLMARK_MITOTIC_SPINDLE	Genes important for mitotic spindle assembly.	*BCL2L11*, *CLIP2*, *ARHGEF2*	3.07 × 10^−2^
HALLMARK_ESTROGEN_RESPONSE_ EARLY	Genes defining early response to estrogen.	*LRIG1*, *ELOVL2*, *MLPH*	3.07 × 10^−2^
HALLMARK_HEME_METABOLISM	Genes involved in metabolism of heme (a cofactor consisting of iron and porphyrin) and erythroblast differentiation.	*NR3C1*, *MXI1*, *SLC6A9*	3.07 × 10^−2^
HALLMARK_INTERFERON_GAMMA_ RESPONSE	Genes up-regulated in response to IFNG	*CDKN1A*, *BTG1*, *UPP1*	3.07 × 10^−2^
HALLMARK_XENOBIOTIC_ METABOLISM	Genes encoding proteins involved in processing of drugs and other xenobiotics.	*ETS2*, *UPP1*, *AKR1C3*	3.07 × 10^−2^

FDR: false discovery rate.

**Table 4 ijms-23-13843-t004:** GSEA based on the hallmark gene sets for downregulated genes of FE25L (p122) compared to FE25 (p19) cells.

Gene Set Name [# Genes (K)]	Description	Genes in Overlap	FDR q-Value
HALLMARK_INTERFERON_GAMMA_ RESPONSE	Genes up-regulated in response to IFNG	*IFI44L*, *MX1*, *TXNIP*, *IFI44*, *USP18*, *IFI27*, *HERC6*, *EPSTI1*, *ISG15*, *CFH*, *IFIT1*, *MX2*, *OAS2*, *OAS3*	4.24 × 10^−15^
HALLMARK_INTERFERON_ALPHA_ RESPONSE	Genes up-regulated in response to alpha-interferon proteins.	*IFI44L*, *MX1*, *TXNIP*, *IFI44*, *USP18*, *IFI27*, *HERC6*, *EPSTI1*, *ISG15*, *IFITM1*, *OAS1*	2.19 × 10^−14^
HALLMARK_EPITHELIAL_ MESENCHYMAL_TRANSI NSITION	Genes defining epithelial-mesenchymal transition, as in wound healing, fibrosis, and metastasis.	*TIMP3*, *THBS1*, *COL3A1*, *VCAN*, *COL4A2*, *IGFBP4*, *TAGLN*, *CDH6*, *COL1A2*, *APLP1*, *NTM*	4.64 × 10^−11^
HALLMARK_COAGULATION	Genes encoding components of blood coagulation system; also up-regulated in platelets.	*CFH*, *TIMP3*, *THBS1*, *OLR1*, *PDGFB*, *MMP15*	1.52 × 10^−5^
HALLMARK_ANGIOGENESIS	Genes are up-regulated during formation of blood vessels (angiogenesis).	*VCAN*, *COL4A2*, *OLR1*	8.58 × 10^−4^
HALLMARK_COMPLEMENT	Genes encode components of the complement system, which is part of the innate immune system.	*CFH*, *COL4A2*, *OLR1*, *PDGFB*, *MMP15*	8.58 × 10^−4^
HALLMARK_ESTROGEN_RESPONSE_ EARLY	Genes defining early response to estrogen.	*IGFBP4*, *CCND1*, *KRT19*, *ITPK1*, *DHRS3*	8.58 × 10^−4^
HALLMARK_MYOGENESIS	Genes involved in development of skeletal muscle (myogenesis).	*COL3A1*, *COL4A2*, *TAGLN*, *ITGB4*, *PTGIS*	8.58 × 10^−4^
HALLMARK_APICAL_ JUNCTION	Genes encoding components of apical junction complex.	*VCAN*, *CDH6*, *ITGB4*, *ACTG2*	6.86 × 10^−3^
HALLMARK_ESTROGEN_RESPONSE_ LATE	Genes defining late response to estrogen.	*IGFBP4*, *CCND1*, *KRT19*, *ITPK1*	6.86 × 10^−3^
HALLMARK_KRAS_SIGNALING_DN	Genes down-regulated by KRAS activation.	*IFI44L*, *MX1*, *SYNPO*, *TENM2*	6.86 × 10^−3^
HALLMARK_ANDROGEN_RESPONSE	Genes defining response to androgens.	*CCND1*, *KRT19*, *ALDH1A3*	8.05 × 10^−3^
HALLMARK_UV_RESPONSE_DN	Genes down-regulated in response to ultraviolet (UV) radiation.	*COL3A1*, *COL1A2*, *CELF2*	2.07 × 10^−2^
HALLMARK_APOPTOSIS	Genes mediating programmed cell death (apoptosis) by activation of caspases.	*TXNIP*, *TIMP3*, *CCND1*	2.62 × 10^−2^
HALLMARK_P53_PATHWAY	Genes involved in p53 pathways and networks.	*TXNIP*, *ITGB4*, *KRT17*	4.39 × 10^−2^

**Table 5 ijms-23-13843-t005:** Sequences of the primers used in this study.

Gene	Sense (5′-3′)	Antisense (5′-3′)	Product Size (bp)
*Ecad*	CGGGAATGCAGTTGAGGATC	AGGATGGTGTAAGCGATGGC	201
*Ncad*	ACCAGGTTTGGAATGGGACA	ACATGTTGGGTGAAGGGGTG	156
*Fibronectin*	CAGTGGGAGACCTCGAGAAG	TCCCTCGGAACATCAGAAAC	168
*Vimentin*	GCATGTCCAAATCGATGTGG	ATTGTTCCGGTTGGCAGCCT	163
*Twist*	GGAGTCCGCAGTCTTACGAG	TCTGGAGGACCTGGTAGAGG	201
*CXCL1*	AACCCCAAGTTAGTTCAATCTGGA	CATGTTGCAGGCTCCTCAGAA	104
*CXCL2*	TCAAACCCAAGTTAGTTCAATCCTGA	GCTGACATGTGATATGTCATCACGAA	113
*IL-6*	TACCCCCAGGAGAAGATTCC	TTTTCTGCCAGTGCCTCTTT	175
*IL-8*	GACATACTCCAAACCTTTCCAC	TTCTCAGCCCTCTTCAAAAAC	180
*BMP2*	AACACTGTGCGCAGCTTCC	CTCCGGGTTGTTTTCCCAC	74
*TIMP1*	TCTGCAATTCCGACCTCGTCATCA	AAGGTGGTCTGGTTGACTTCTGGT	67
*IGF2*	ACACCCTCCAGTTCGTCTGT	GGGGTATCTTGGGGAAGTTGT	213
*VEGFA*	CTACCTCCACCATGCCAAGT	GCAGTAGCTGCGCTGATAGA	109
*CDKN2A*	GTGGACCTGGCTGAGGAG	CTTTCAATCGGGGATGTCTG	132
*SFRP1*	TTGAGCATTTGAAAGGTGTGCTA	ACAGCTACACTACCAGGGAAATCC	121
*FRZB*	CCTGCCCTGGAACATGACTAA	CAGACCTTCGAACTGCTCGAT	91
*SFRP4*	TGTGTTACGAGTGGCG	GGGGGATTACTACGACTG	172
*CCND1*	TCCTCTCCAAAATGCCAGAG	GGCGGATTGGAAATGAACTT	109
*IKKe*	ACTCTGGAAGTGGCAAGGACAT	TACCTGATCCCGGCTCTTCACCA	234
*S100A4*	AGTTCAAGCTCAACAAGTCAGAACTAA	TCATCTGTCCTTTTCCCCAAGA	79
*SPP1*	GCCAGTTGCAGCCTTCTCA	AAAAGCAAATCACTGCAATTCTCA	75
*C-MYC*	CCTGGTGCTCCATGAGGAGAC	CAGACTCTGACCTTTTGCCAGG	128
*GNAL*	CACCAGATGCAGGAGAAGATCC	AGGTGAAGTGCGGGTAGCAGTA	120
*TMEM130*	GCTCCTATCTCACTAAGACCGTC	CACGGAGTCTTCAGTCACCATC	134
*DNMT3B*	TAACAACGGCAAAGACCGAGGG	TCCTGCCACAAGACAAACAGCC	110
*CCND1*	TCCTCTCCAAAATGCCAGAG	GGCGGATTGGAAATGAACTT	109
*CASP1*	GCTGAGGTTGACATCACAGGCA	TGCTGTCAGAGGTCTTGTGCTC	145
*EHD1*	GCGTTTGGCAACGCTTTCCTCA	ATCCGCTGCTTCTCTCCAGACA	116
*P2RY2*	CGAGGACTTCAAGTACGTGCTG	GTGGACGCATTCCAGGTCTTGA	123
*GAPDH*	GGTCTCCTCTGACTTGAACA	GTGAGGGTCTCTCTCTTCCT	221

## Data Availability

The datasets generated during and analyzed during the current study are available from the corresponding author on reasonable request.

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
