# Peer review of "Spontaneous Transformation of a p53 and Rb-Defective Human Fallopian Tube Epithelial Cell Line after Long Passage with Features of High-Grade Serous Carcinoma"

_ijms, 2022, doi:10.3390/ijms232213843_

Round 1

Reviewer 1 Report (Previous Reviewer 4)

All the issues have been corrected

Author Response

All the issues have been corrected

Response: Thanks for your comments.

Reviewer 2 Report (Previous Reviewer 3)

Major comments:

The authors’ responses to reviewers’ comments are appreciated.

Further clarifications and better presentations are needed to support the claim that the proposed cell line can serve as suitable model of high grade serous ovarian carcinoma, which has distinct histologic and somatic mutational profiles.

(1) Figure 4. The high-power photos of tumors do show high grade features, but the cytology is non-specific for its classification. What did the architecture and morphology of the tumors look like, and how did those compare to high grade serous carcinoma in particular?

(2) The tumors generated from FE25L cells showed high genetic instability as would have been expected in several other subtypes of high grade carcinomas or malignant neoplasia. Clarifications are needed to illustrate how closely the molecular profile reported in HGSOC in particular compares with that of tumors generated with FE25L and FE25 cells. The authors stated that “…using the top 500 diverse genes of Group 1 HGSOC, FE25 and FE25L cells were grouped with Group 1 cells…Fig. 10)”(lines 257-259). There appears lack of statistical measures cited/highlighted in figure 10 on how closely FE25 and/or FE25L cells correlated and clustered with the group I cell lines vs other cell lines shown in the figure.

(3). Statistical analyses: If means are being presented in graphs then standard deviations (SDs) instead of SEM should be shown to display the actual observed data spread. In figure 1 and figure 7, SEM is noted to be presented which may not be appropriate.

Minor comments:

There are still multiple typographical/font issues that should be fixed throughout the manuscript:

For example:

-“…Trp53 and Rb…” (line 286)” gene names should be in italics by convention.

-“…on days 3 and day5”(lines 80-81):  should be “on day 3 and day 5”

-Inconsistent font sizes at multiple places (e.g. lines 148-152)  in the manuscript including the main text and figure legends (e.g. legend for figure 3).

…etc.

Author Response

Major comments:

The authors’ responses to reviewers’ comments are appreciated.

Further clarifications and better presentations are needed to support the claim that the proposed cell line can serve as suitable model of high grade serous ovarian carcinoma, which has distinct histologic and somatic mutational profiles.

(1) Figure 4. The high-power photos of tumors do show high grade features, but the cytology is non-specific for its classification. What did the architecture and morphology of the tumors look like, and how did those compare to high grade serous carcinoma in particular?

Response: We are sorry the gross picture of the xenograft tumor was not available. We added a gross picture of xenografts in the wax block (Figure 4B). In the IHC of HGSOC, PAX8 and WT1 expressions are the key features (Cybula et al. 2021). In Figure 4D, the xenografts revealed positive staining of PAX8 and WT1, which is consistent with the expression pattern of HGSOC.  

Cybula, Magdalena, Lin Wang, Luyao Wang, Ana Luiza Drumond-Bock, Katherine M. Moxley, Doris M. Benbrook, Camille Gunderson-Jackson, et al. 2021. “Patient-Derived Xenografts of High-Grade Serous Ovarian Cancer Subtype as a Powerful Tool in Pre-Clinical Research.” Cancers 13 (24):6288

(2) The tumors generated from FE25L cells showed high genetic instability as would have been expected in several other subtypes of high grade carcinomas or malignant neoplasia. Clarifications are needed to illustrate how closely the molecular profile reported in HGSOC in particular compares with that of tumors generated with FE25L and FE25 cells. The authors stated that “…using the top 500 diverse genes of Group 1 HGSOC, FE25 and FE25L cells were grouped with Group 1 cells…Fig. 10)”(lines 257-259). There appears lack of statistical measures cited/highlighted in figure 10 on how closely FE25 and/or FE25L cells correlated and clustered with the group I cell lines vs other cell lines shown in the figure.

Response: We added another figure to illustrate the most correlated ovarian cancer cell line to FE25 and FE25L after comparing with 500 genes (Figure 10B). We also added a paragraph to discuss the results.  

(3). Statistical analyses: If means are being presented in graphs then standard deviations (SDs) instead of SEM should be shown to display the actual observed data spread. In figure 1 and figure 7, SEM is noted to be presented which may not be appropriate.

Response: We changed the statistic method to SD. 

Minor comments:

There are still multiple typographical/font issues that should be fixed throughout the manuscript:

For example:

-“…Trp53 and Rb…” (line 286)” gene names should be in italics by convention.

Response: We revised it accordingly. 

-“…on days 3 and day5”(lines 80-81):  should be “on day 3 and day 5”

Response: We revised it accordingly. 

-Inconsistent font sizes at multiple places (e.g. lines 148-152)  in the manuscript including the main text and figure legends (e.g. legend for figure 3).

…etc.

Response: We unified the font size in the manuscript.

Reviewer 3 Report (Previous Reviewer 2)

Ovarian cancer is a lethal cancer-type and despite the improvement of clinical treatments, unfortunately it is still associated with a poor survival rate. Chang et al., in this manuscript aimed to compare previous characterized human fallopian tube epithelial cell line, FE25 with p53 and Rb deficiencies, in culture for low (passage 17–31) and high passages (passage 120–139) in order to determine whether these cells may serve as a model for ovarian cancer high-grade serous carcinomas (HGSOC). The authors found that compared to low passage number cells, high passage number cells, FE25L, showed increased cell proliferation, clonogenicity, polyploidy, aneuploidy, cell migration, and invasion, as well as the ability to grow tumors in xenograft. RNA-seq analysis, qRT-PCR and cross-platform analysis corroborated that FE25L cells showed a more aggressive malignant behavior than FE25 cells and consequently may provide a suitable model for HGSOC research.

The manuscript is well written and organized. Minor comments are found below.

-Pag.2, line 65: It is not clear the difference between FTE and FTEC. Please, clarify it.

-Fig. 1A: The scale bar of the Figure does not match the one described in the legend. Please modify it.

-Fig. 3: Please, check the font of the Figure legend.

-Pag. 5, line 122: Please, open H&E.

-Fig. 4A: What does it mean Fe25 on the top of the graph? Please, clarify it.

General comment:

-Please, check the font of the manuscript since there are many parts that would need to be formatted, as for example section 2.6.

Author Response

Ovarian cancer is a lethal cancer-type and despite the improvement of clinical treatments, unfortunately it is still associated with a poor survival rate. Chang et al., in this manuscript aimed to compare previous characterized human fallopian tube epithelial cell line, FE25 with p53 and Rb deficiencies, in culture for low (passage 17–31) and high passages (passage 120–139) in order to determine whether these cells may serve as a model for ovarian cancer high-grade serous carcinomas (HGSOC). The authors found that compared to low passage number cells, high passage number cells, FE25L, showed increased cell proliferation, clonogenicity, polyploidy, aneuploidy, cell migration, and invasion, as well as the ability to grow tumors in xenograft. RNA-seq analysis, qRT-PCR and cross-platform analysis corroborated that FE25L cells showed a more aggressive malignant behavior than FE25 cells and consequently may provide a suitable model for HGSOC research.

The manuscript is well written and organized. Minor comments are found below.

-Pag.2, line 65: It is not clear the difference between FTE and FTEC. Please, clarify it.

Response: We changed FTE to FTEC to avoid confusion. 

-Fig. 1A: The scale bar of the Figure does not match the one described in the legend. Please modify it.

Response: We modify it to 1000 μm. 

-Fig. 3: Please, check the font of the Figure legend.

Response: We changed the font of the Figure legend to size 9. 

-Pag. 5, line 122: Please, open H&E.

Response: We spelled out H & E. 

-Fig. 4A: What does it mean Fe25 on the top of the graph? Please, clarify it.

Response: We changed the title to FE25. 

General comment:

-Please, check the font of the manuscript since there are many parts that would need to be formatted, as for example section 2.6.

Response: We changed the font to size 10 in section 2.6. We also modified the font size of the other parts.

This manuscript is a resubmission of an earlier submission. The following is a list of the peer review reports and author responses from that submission.

Round 1

Reviewer 1 Report

This manuscript is a thorough description of the cell line FE25/FE25L that was developed by the authours from primary FT epithelium. The acquisition of additional genomic perturbation and mutations in cell lines that have been passaged for a long time is a well known phenomenom, undermining the utility of cell lines as research tools and the ability to reproduce research results. Nonetheless, the paucity of models of HGSOC makes this manuscript useful. 

1. The methods that were used in this study are "classic", including - XTT, soft agar assay, migration and invasion, Karyotyping. The ability to assess the soundness of the results and the significance of the differences between the 2 cell line populations depends heavily on the number of repeats of each experiments. This is not mentioned in the motheds section. In some of the figures the differences seem rather marginal, while the authors claim statistical significance - The methodology is concerning. 

2. Section 4.10 - It is unclear which method was used for transcriptomics - microarray or RNA seq. 

3. Figure 10 - the autors claim that it reflects clustering of FE25 with HGSOC cell lines, however this conclusion is not supported by the un-supervised clustering seen in this heat-map. The explenation of the mothods in 4.11 is too vague to understand what was done. 

4. The methods sections are too detailed and in most cases can be replaced by a reference or the manufacturer protocols.

5. The potential of forming xenografts was tested in 3 mice per group and the period of time until clinically evident tumor was very long. Conclusions cannot be drawn from such a small cohort size. 

6. Publications by Crum and Drapkin were not cited although they were the pioneers in this field. 

Author Response

Reviewer 1

This manuscript is a thorough description of the cell line FE25/FE25L that was developed by the authours from primary FT epithelium. The acquisition of additional genomic perturbation and mutations in cell lines that have been passaged for a long time is a well known phenomenom, undermining the utility of cell lines as research tools and the ability to reproduce research results. Nonetheless, the paucity of models of HGSOC makes this manuscript useful. 

  1. The methods that were used in this study are "classic", including - XTT, soft agar assay, migration and invasion, Karyotyping. The ability to assess the soundness of the results and the significance of the differences between the 2 cell line populations depends heavily on the number of repeats of each experiments. This is not mentioned in the motheds section. In some of the figures the differences seem rather marginal, while the authors claim statistical significance - The methodology is concerning. 

Response: We added the numbers of experiments in each figure legend. 

  1. Section 4.10 - It is unclear which method was used for transcriptomics - microarray or RNA seq. 

Response: We used RNA seq. 

  1. Figure 10 - the autors claim that it reflects clustering of FE25 with HGSOC cell lines, however this conclusion is not supported by the un-supervised clustering seen in this heat-map. The explenation of the mothods in 4.11 is too vague to understand what was done. 

Response: We added more detailed methods in 4. 11. 

  1. The methods sections are too detailed and in most cases can be replaced by a reference or the manufacturer protocols.

Response: We shortening the methods section. 

  1. The potential of forming xenografts was tested in 3 mice per group and the period of time until clinically evident tumor was very long. Conclusions cannot be drawn from such a small cohort size. 

Response: We modified the conclusion to a lower tone. 

  1. Publications by Crum and Drapkin were not cited although they were the pioneers in this field. 

Response: We added their studies in the introduction section (Karst, Levanon, and Drapkin 2011; Karst et al. 2014)

Karst, Alison M., Paul M. Jones, Natalie Vena, Azra H. Ligon, Joyce F. Liu, Michelle S. Hirsch, Dariush Etemadmoghadam, David D. L. Bowtell, and Ronny Drapkin. 2014. “Cyclin E1 Deregulation Occurs Early in Secretory Cell Transformation to Promote Formation of Fallopian Tube-Derived High-Grade Serous Ovarian Cancers.” Cancer Research 74 (4): 1141–52.

Karst, Alison M., Keren Levanon, and Ronny Drapkin. 2011. “Modeling High-Grade Serous Ovarian Carcinogenesis from the Fallopian Tube.” Proceedings of the National Academy of Sciences of the United States of America 108 (18): 7547–52.

Author Response

Reviewer 2

Ovarian cancer is a lethal cancer-type and despite the improvement of clinical treatments, unfortunately it is still associated with a poor survival rate. Chang et al., in this manuscript aimed to compare previous characterized human fallopian tube epithelial cell line, FE25 with p53 and Rb deficiencies, in culture for low (passage 17–31) and high passages (passage 120–139) in order to determine whether these cells may serve as a model for ovarian cancer high-grade serous carcinomas (HGSOC). The authors found that compared to low passage number cells, high passage number cells, FE25L, showed increased cell proliferation, clonogenicity, polyploidy, aneuploidy, cell migration, and invasion, as well as the ability to grow tumors in xenografts. RNA-seq analysis and qRT-PCR corroborated that FE25L cells showed a more aggressive malignant behavior than FE25 cells and consequently may provide a suitable model for HGSOC research.

The manuscript is well written and organized. Minor comments are found below.

For each Figure, I would suggest including:

- same font labeling

- the passage number of the cells

- the number of experiments and biological replicates used

Response:  We revised them according to the suggestions. 

Fig.1: The authors should provide with more information about the number of experiments and biological replicates used to summarize the results in this Figure. Moreover, the aim of the study was to use cells with different passage numbers, but the Figure legend at pag. 3, line 84 would suggest that the cells have been grown only for 5 days. Please, clarify. The same concern arises for the results in Fig. 2.

Response: We added passages of cells in the figure legend. The proliferation was tested on days 1, 3, and 5. The migration and invasion assay was done for 48 hours.  

Fig. 1A: What does represent the green element within the panel? Sorry, I missed it!

Response: The green words and line indicated a 1000 μm scale. 

Fig. 1B: The authors should indicate cell proliferation using a log-scale or a fold-change graph.

Response: It was a fold-change graph. 

Fig.1C: The distribution of the cells through G2/M phase is a concern here. Please, adjust the gating and the cell distribution.

Response: We revised the figure. 

Fig.2A and B: The images appear to be too much manipulated. I would suggest to use the original/unmodified images with the crystal violet. What does it mean “fold cell number” on the Y axis of the graphs? Why is the scale bar different? I would suggest to use more consistent labeling.

Response: We revised the figure. The cell number of FE25 in three experiments got a mean value. Then we compared each FE25 and FE25L cell number with the mean value. Three values of folds of FE25 and FE25L were compared and analyzed. We unified the scale bar of the Y axis. 

Fig. 3, pag.5, line 113: Could the authors include more information about the quantification method?

Response: The same analytical method used in Figure 2 was applied. We added the information in Figure 3. The colony number was measured by ImageJ. 

Fig. 3, pag. 5, line 114: As for Fig. 1. The authors should provide with more information about the number of experiments and biological replicates used to summarize the results in this Figure.

Response: We added the information (n=3 in each group) in the figure legend. 

Again, the aim of the study was to use cells with different passage numbers, but the cells have been grown for 14 days. I would suggest including also the passage number of the cells.

Response: The passage number of cells was added in the figure legend. 

Fig.7: “RQ” in the Y axis does not mean anything in all the panels. Please, modify the labeling.

Response: We changed RQ to “Relative quantification of GAPDH”

Fig. 8B-D: Do the authors mean R.Q. of beta-catenin (relative to beta-actin)? Please, modify. How many time the WB has been performed? How the bands quantified? Please, add more information.

Response: We revised RQ in the figure. Three times WB has been performed. The quantification method (we used ImageJ software) was described in the method section. 

Fig. 10: What is the biological/pathological significance of these genes?

Response: Figure 10 shows a list of high-grade serous ovarian carcinoma (HGSOC) cell lines. We analyzed the genetic background of these cancer cell lines and found FE25 and FE25L owned similar genetic expressions with several HGSOC cell lines.   

Pag.9, line 177: Could the authors justify the choice to focus on 100  up-regulated/down-regulated genes?

Response: We thought 100 up-regulated/down-regulated genes of FE25L compared with FE25 were the prominent genes in the phenotype. 

Table 4 is very busy and without relative legend. To make the table more informative, I would suggest using heat-maps and enrichment signaling pathways analysis.

The same for Table 5.

Response: Due to only FE25L compared to FE25, the heat maps only one column, which was unsuitable for data presentation.  

Material and Methods:

I would suggest to combine 4.14 with 4.15 since it belongs to WB analysis.

Please move this section at the end, after “Conclusions”.

Response: We combined 4.14 and 4.15 and moved the methods section after “conclusion”.

Reviewer 3 Report

Chang et al aimed to compare FE25 cells cultured in vitro for low and high passages to determine whether the cell lines can serve as suitable models of high grade serous ovarian carcinoma (HGSOC) by comparing their in vitro performance, genomic alterations and ability to grow in xenografts.

Major comments:

(1) The study aim has clear relevance to basic and translational research related to HGSOC. The study employs appropriate assays to examine the experimental utility of the tested cell lines by evaluating cell proliferation, aneuploidy, abilities of cell migration and invasion, resistance to chemotherapy, ability to grow tumors in xenografts, and their genomic profiles.

(2) Further clarifications are needed on the methods to better support the study analyses and results:

(2.1). Additional information is needed to support study reproducibility. It is unclear in the manuscript whether and how many replicates and/or independent experiments were performed to obtain data for each part of the study, and how reproducible the different numbers of passages were in yielding the presented findings.

(2.2). TP53 genetic alteration is a main driver event in the pathogenesis of HGSOC. It is noted in the text that the cell line being evaluated was obtained by immortalizing human fallopian tubal epithelial cells by transfection of human papillomavirus 16 E6/E7 genes, resulting in downregulation of TP53 and RB genes. Data or figure from the experiments performed to confirm the TP53 and RB status of these cells should be provided in at least the supplemental section to verify the efficiency of the transfection.

(2.3). Xenografting experiments: Lines 482-483: “We observed and measured the tumor growth in each group of mice. Tumor volume was calculated as …” Clarification is needed on how the tumor volume was examined, e.g. by palpation only, or with imaging/bioluminescent imaging, or other measurement aid(s).

(2.4). Histology of tumor developed from tested cell lines: More information is needed on describing whether these tumors morphologically resembled HGSOC in terms of architecture and pathologic features. Other than PAX8 and WT-1, expression of other characteristic protein(s) related to HGSOC development, e.g. stathmin, CK7, would also be of interest in this setting to see how well these cells recapitulated human HGSOC.

(2.5). Text lines 234-235: “FE25L cells were found to be more sensitive to chemotherapeutic drugs because more mutations were noted.” The result and figure sections appear to focus mostly on gene expression analyses with lack of information on the somatic mutational profile related to these cells. More details are needed on how the mutational profile reported in human HGSOC compares with that of FE25L cells.

(2.6). A clearer presentation of a correlation index with human HGSOC genomic changes and CNVs is also needed to further support FE25L being a suitable model for studying HGSOC.

(2.7). Page 12 of 24: The authors showed expression of 5 select genes related to the EMT. It is unclear based on the manuscript how these 5 genes were selected, if other EMT-related genes or transcription factors were also examined and whether the same pattern was seen with these other genes, given that there are multiple/ at least 30+ EMT-related genes or transcription factors (according to the consensus statement on EMT led by the EMT International Association; Nat Rev Mol Cell Biol. 2020;21(6):341-352). 

(2.8). Statistical analyses: Non-parametric tests and presentation of medians and interquartile ranges may be more appropriate than using t-tests and ANOVA unless the normality assumption for these tests is justified with the data. If means are being presented in graphs then standard deviations (SDs) instead of SEM should be shown to display the actual observed data spread.

Minor comments:

(1). Text, line 122 “FE25L tumors showed higher WT1 expression”. Clarification is needed to specify if that refers to percentage of tumor cells being positive for WT-1.

(2). Text, line 498-499:  Clarification is needed to specify whether that refers to intensity of staining, percentage of tumor cells showing positive expression, or a combined measure of both, as these are all different parameters used in evaluating immunohistochemical expression.

(3). Figures 6A and 6B should be better labeled with regard to FE25 cells or FE25L cells, rather than numbers “p20” “p124”.

(4). Multiple typographical/grammatical/font issues should be fixed for the revision:

Some examples for consideration (not all listed):

All gene names, e.g. TP53, BRCA1, ERBB2 …etc, should be italicized in the text by convention.

Text, line 225: “nucleus” should be in plural form - nuclei.

Text, line 329: “obtained for Dr. Chu TY’s lab” should be “obtained from…” the lab providing the materials

Text, line 333: “they were passage” should be “they were passaged”

Superscripts and subscripts should be checked and applied as appropriate:

Some examples for consideration (not all listed):

Text, line 117: “1 × 106” should be 1 × 106

Text, line 331: “5 × 105” should be 5x105

Text, line 332: 75 cm2 should be 75 cm2

Text, line 332: CO2 should be CO2       

…etc.

Author Response

Reviewer 3. 

Chang et al aimed to compare FE25 cells cultured in vitro for low and high passages to determine whether the cell lines can serve as suitable models of high grade serous ovarian carcinoma (HGSOC) by comparing their in vitro performance, genomic alterations and ability to grow in xenografts.

Major comments:

(1) The study aim has clear relevance to basic and translational research related to HGSOC. The study employs appropriate assays to examine the experimental utility of the tested cell lines by evaluating cell proliferation, aneuploidy, abilities of cell migration and invasion, resistance to chemotherapy, ability to grow tumors in xenografts, and their genomic profiles.

(2) Further clarifications are needed on the methods to better support the study analyses and results:

(2.1). Additional information is needed to support study reproducibility. It is unclear in the manuscript whether and how many replicates and/or independent experiments were performed to obtain data for each part of the study, and how reproducible the different numbers of passages were in yielding the presented findings.

Response: We added the information on the number of replicates in the figure legends. 

(2.2). TP53 genetic alteration is a main driver event in the pathogenesis of HGSOC. It is noted in the text that the cell line being evaluated was obtained by immortalizing human fallopian tubal epithelial cells by transfection of human papillomavirus 16 E6/E7 genes, resulting in downregulation of TP53 and RB genes. Data or figure from the experiments performed to confirm the TP53 and RB status of these cells should be provided in at least the supplemental section to verify the efficiency of the transfection.

Response: This information has been published in our previous study (Huang et al. 2015)

Huang, Hsuan-Shun, Sung-Chao Chu, Che-Fang Hsu, Pao-Chu Chen, Dah-Ching Ding, Meng-Ya Chang, and Tang-Yuan Chu. 2015. “Mutagenic, Surviving and Tumorigenic Effects of Follicular Fluid in the Context of p53 Loss: Initiation of Fimbria Carcinogenesis.” Carcinogenesis 36 (11): 1419–28.

(2.3). Xenografting experiments: Lines 482-483: “We observed and measured the tumor growth in each group of mice. Tumor volume was calculated as …” Clarification is needed on how the tumor volume was examined, e.g. by palpation only, or with imaging/bioluminescent imaging, or other measurement aid(s).

Response: Tumor was detected by palpation only.

(2.4). Histology of tumor developed from tested cell lines: More information is needed on describing whether these tumors morphologically resembled HGSOC in terms of architecture and pathologic features. Other than PAX8 and WT-1, expression of other characteristic protein(s) related to HGSOC development, e.g. stathmin, CK7, would also be of interest in this setting to see how well these cells recapitulated human HGSOC.

Response: We added CK7 to the figure. We added more descriptions of the morphology of xenograft tumors that resembled HGSOC. 

(2.5). Text lines 234-235: “FE25L cells were found to be more sensitive to chemotherapeutic drugs because more mutations were noted.” The result and figure sections appear to focus mostly on gene expression analyses with lack of information on the somatic mutational profile related to these cells. More details are needed on how the mutational profile reported in human HGSOC compares with that of FE25L cells.

Response: The mutation profile was only evaluated by CNV. We may add the mutational profile in future studies. 

(2.6). A clearer presentation of a correlation index with human HGSOC genomic changes and CNVs is also needed to further support FE25L being a suitable model for studying HGSOC.

Response: The correlation of genomic change between HGSOC and FE25L was illustrated in Figure 10. The correlation of CNV between FE25L and HGSOC may be evaluated in the future. 

(2.7). Page 12 of 24: The authors showed expression of 5 select genes related to the EMT. It is unclear based on the manuscript how these 5 genes were selected, if other EMT-related genes or transcription factors were also examined and whether the same pattern was seen with these other genes, given that there are multiple/ at least 30+ EMT-related genes or transcription factors (according to the consensus statement on EMT led by the EMT International Association; Nat Rev Mol Cell Biol. 2020;21(6):341-352). 

Response: We did not evaluate other EMT-related genes. We added the reference in the discussion section. 

(2.8). Statistical analyses: Non-parametric tests and presentation of medians and interquartile ranges may be more appropriate than using t-tests and ANOVA unless the normality assumption for these tests is justified with the data. If means are being presented in graphs then standard deviations (SDs) instead of SEM should be shown to display the actual observed data spread.

Response: The calculation was counted using a non-parametric test. We used SD instead of SEM in the figure. The data was calculated again. 

Minor comments:

(1). Text, line 122 “FE25L tumors showed higher WT1 expression”. Clarification is needed to specify if that refers to percentage of tumor cells being positive for WT-1.

Response: We added the percentage of WT-1 positive cells in the text. 

(2). Text, line 498-499:  Clarification is needed to specify whether that refers to intensity of staining, percentage of tumor cells showing positive expression, or a combined measure of both, as these are all different parameters used in evaluating immunohistochemical expression.

Response: We counted positive staining cells in 50 cells counted in each of three fields which was similar to the percentage method. We clarified this method in the method section. 

(3). Figures 6A and 6B should be better labeled with regard to FE25 cells or FE25L cells, rather than numbers “p20” “p124”.

Response: We changed them to FE25 and FE25L.

(4). Multiple typographical/grammatical/font issues should be fixed for the revision:

Some examples for consideration (not all listed):

All gene names, e.g. TP53, BRCA1, ERBB2 …etc, should be italicized in the text by convention.

Text, line 225: “nucleus” should be in plural form - nuclei.

Text, line 329: “obtained for Dr. Chu TY’s lab” should be “obtained from…” the lab providing the materials

Text, line 333: “they were passage” should be “they were passaged”

Superscripts and subscripts should be checked and applied as appropriate:

Some examples for consideration (not all listed):

Text, line 117: “1 × 106” should be 1 × 106

Text, line 331: “5 × 105” should be 5x105

Text, line 332: 75 cm2 should be 75 cm2

Text, line 332: CO2 should be CO2       

…etc.

Response: We corrected all the errors.

Reviewer 4 Report

The authors present a comprehensive analysis of how culture passages number affects  carcinogenesis by comparing low-passage cells (FE25) and high-passage cells (FE25L) and whether the latter can mimic a cellular model of HGSOC.

I have the following comments:

The study replicate and confirms that the more culture passages the cells have, the more aggressive cancer behavior is shown. Nevertheless, despite an exhaustive description of cancer hallmarks in the results section, the discussion seems more like a summary of their findings, it would be appreciated to introduce more references to other investigations carried out before, to further argue in discussion section.

In Introduction line 54  “TP53 mutations are found in more than 96% of HGSOC cases”  it should be clarified if authors refer to somatic of germline mutations, the same in line 57 “Approximately half of HGSOC also harbour mutations…” somatic, germline of both?

In line 154 please provide reference to “Compared to FE25, FE25L had gain sites at chr2p25, 7q21,22, 8p11, and 20q11, which are related to breast cancer and chronic myelogenous leukemia”

Some minor issues that should be addressed:

-        Gene names should be in italics, please correct it

-        Uniform style in unit’s nomenclature, sometimes the authors write ml or mL.

-        Abbreviations: In line 64 FTEC stands for? and in line 463 PBST stands for?

Author Response

Reviewer 4. 

The authors present a comprehensive analysis of how culture passages number affects  carcinogenesis by comparing low-passage cells (FE25) and high-passage cells (FE25L) and whether the latter can mimic a cellular model of HGSOC.

I have the following comments:

The study replicate and confirms that the more culture passages the cells have, the more aggressive cancer behavior is shown. Nevertheless, despite an exhaustive description of cancer hallmarks in the results section, the discussion seems more like a summary of their findings, it would be appreciated to introduce more references to other investigations carried out before, to further argue in discussion section.

Response: We added a paragraph to discuss previously published studies. 

In Introduction line 54  “TP53 mutations are found in more than 96% of HGSOC cases”  it should be clarified if authors refer to somatic of germline mutations, the same in line 57 “Approximately half of HGSOC also harbour mutations…” somatic, germline of both?

Response: We added “somatic” before “mutation”.  Both somatic and germline mutations were noted in the last sentence.

In line 154 please provide reference to “Compared to FE25, FE25L had gain sites at chr2p25, 7q21,22, 8p11, and 20q11, which are related to breast cancer and chronic myelogenous leukemia”

Response: We added two references on it (Voutsadakis 2020; Al-Achkar et al. 2012)

Al-Achkar, Walid, Abdulsamad Wafa, Faten Moassass, and Thomas Liehr. 2012. “A Chronic Myeloid Leukemia Case with a Unique Variant Philadelphia Translocation: t(9;22;21)(q34;q11;p12).” Oncology Letters 3 (5): 1027–29.

Voutsadakis, Ioannis A. 2020. “8p11.23 Amplification in Breast Cancer: Molecular Characteristics, Prognosis and Targeted Therapy.” Journal of Clinical Medicine 9 (10): 3079.

Some minor issues that should be addressed:

-        Gene names should be in italics, please correct it

Response: We correct gene names in italics. 

-        Uniform style in unit’s nomenclature, sometimes the authors write ml or mL.

Response: We changed “ml” to “mL”. 

-        Abbreviations: In line 64 FTEC stands for? and in line 463 PBST stands for?

Response: We spell out the FTEC (fallopian tube epithelial cell). PBST (PBS Tween-20 Buffer)

Round 2

Reviewer 1 Report

Still missing references to Crum and Drapkin's work, such as : PMID: 17218844 or PMID: 18854563

I expect a work that is entirely based on the description of a single cell line to validate the results with more than 3 technical repeats. In my opinion, a more meaningful validation should be by using at least 3 different cell batches from different number of passages in each experiment. 

Section 5.11 - I still don't understand the sentence "The top 500 diverse genes of 528 group 1 were selected to differentiate the difference between the cell lines.". 

Author Response

Still missing references to Crum and Drapkin's work, such as : PMID: 17218844 or PMID: 18854563

Response: We added these two references in the introduction. 

I expect a work that is entirely based on the description of a single cell line to validate the results with more than 3 technical repeats. In my opinion, a more meaningful validation should be by using at least 3 different cell batches from different number of passages in each experiment. 

Response: Thanks for the suggestion. The FE25 cell line has been extensively studied in our several previous publications [1–7]. The malignant transformation phenotypes of this cell line have been repeated studies in different cell batches with different passage numbers. They showed consistent results. Therefore, we believed the current results could be trusted.  

[1] Huang H-S, Hsu C-F, Chu S-C, Chen P-C, Ding D-C, Chang M-Y, et al. Haemoglobin in pelvic fluid rescues Fallopian tube epithelial cells from reactive oxygen species stress and apoptosis. J Pathol 2016;240:484–94.

[2] Huang H-S, Chen P-C, Chu S-C, Lee M-H, Huang C-Y, Chu T-Y. Ovulation sources coagulation protease cascade and hepatocyte growth factor to support physiological growth and malignant transformation. Neoplasia 2021;23:1123–36.

[3] Hsu C-F, Chen P-C, Seenan V, Ding D-C, Chu T-Y. Ovulatory Follicular Fluid Facilitates the Full Transformation Process for the Development of High-Grade Serous Carcinoma. Cancers 2021;13:468.

[4] Huang H-S, Chu S-C, Hsu C-F, Chen P-C, Ding D-C, Chang M-Y, et al. Mutagenic, surviving and tumorigenic effects of follicular fluid in the context of p53 loss: initiation of fimbria carcinogenesis. Carcinogenesis 2015;36:1419–28.

[5] Wu N-Y, Huang H-S, Chao TH, Chou HM, Fang C, Qin C-Z, et al. Progesterone Prevents High-Grade Serous Ovarian Cancer by Inducing Necroptosis of p53-Defective Fallopian Tube Epithelial Cells. Cell Rep 2017;18:2557–65.

[6] Hsu C-F, Huang H-S, Chen P-C, Ding D-C, Chu T-Y. IGF-axis confers transformation and regeneration of fallopian tube fimbria epithelium upon ovulation. EBioMedicine 2019;41:597–609.

[7] Wang K-H, Chu S-C, Chu T-Y. Loss of calponin h1 confers anoikis resistance and tumor progression in the development of high-grade serous carcinoma originating from the fallopian tube epithelium. Oncotarget 2017;8:61133–45.

Section 5.11 - I still don't understand the sentence "The top 500 diverse genes of 528 group 1 were selected to differentiate the difference between the cell lines.". 

Response: In the revised version, there was no “528” in the text. 

Reviewer 2 Report

The authors have addressed the majority of my concerns. However, Figure 1C is still not correct being the gate of G2/M wrong. This Figure needs to be removed or up-dated.